# Theoretical Analysis of Relative Errors in Gradient Computations for Adversarial Attacks with CE Loss

## Abstract

Gradient-based adversarial attacks using the Cross-Entropy (CE) loss often over-estimate robustness due to relative errors in gradient computation induced by floating-point arithmetic. Empirical methods like MIFPE mitigate this by scaling logits with a factor $c = T/\Delta_{\text{detach}}$ where $T = 1$, significantly improving evaluation accuracy. However, a theoretical understanding of these errors remains limited. To bridge this gap, we pioneer the first rigorous theoretical analysis of floating-point errors in CE-based gradient attacks, systematically dissecting relative errors across four distinct scenarios: (i) unsuccessful untargeted attacks, (ii) successful untargeted attacks, (iii) unsuccessful targeted attacks, and (iv) successful targeted attacks. This foundational study uncovers novel patterns in numerical instability and derives the optimal scaling factor $T = t^*$ that minimizes error impact in each scenario. Notably, our analysis reveals that $t^*$ closely approximates 1 in unsuccessful untargeted attacks, providing a theoretical justification for MIFPE's empirical choice and addressing prior optimality gaps. To validate the correctness of our theoretical derivations, we refine MIFPE by incorporating $T = t^*$ into the Theoretical MIFPE (T-MIFPE) loss function, which further reduces floating-point-induced errors. Comprehensive experiments validate our theory.

## 1 Introduction

Deep learning has revolutionized artificial intelligence, powering breakthroughs in safety-critical domains such as aviation Le Clainche et al. (2023), medical diagnosis Yadav and Jadhav (2019), and autonomous driving Feng et al. (2020). Its pervasive adoption amplifies societal benefits while introducing profound risks. A single imperceptible perturbation to inputs—known as adversarial examples—can mislead models into catastrophic errors Szegedy et al. (2014); Goodfellow et al. (2015), potentially causing accidents in autonomous vehicles Boloor et al. (2020) or misdiagnoses in healthcare. As deep learning permeates interconnected systems, accurate robustness evaluation becomes imperative to ensure safe deployment.

To address these challenges, the research community has proposed numerous evaluation techniques Madry et al. (2018); Shafahi et al. (2019); Alayrac et al. (2019); Zhang et al. (2019); Pang et al. (2020); Wang et al. (2020); Wu et al. (2020b;a); Yu et al. (2021); Gao et al. (2022); Yu et al. (2023). A canonical example is the Projected Gradient Descent (PGD) attack Madry et al. (2018), which exploits gradient information to generate adversarial examples for model robustness assessment. However, studies Croce and Hein (2020a); Mao et al. (2021); Yu et al. (2021) demonstrate that PGD paired with conventional cross-entropy (CE) loss frequently overestimates model robustness. This occurs because CE-computed gradients inadequately guide adversarial example generation—a phenomenon associated with gradient masking Goodfellow (2018). Advanced evaluation methods have thus employed attack algorithm ensembles Croce and Hein (2020a); Mao et al. (2021) to combine multiple strategies for improved accuracy. Yet these methods lack fundamental analysis of root causes behind gradient-based attack overestimation. Research by Gao et al. (2022) reveals persistent overestimation issues even when evaluating defenses like ensemble methods Kariyappa and Qureshi (2019); Pang et al. (2019); Yang et al. (2020).

The failure of gradient-based attacks like PGD with CE loss has prompted the development of alternative loss functions to mitigate the overestimation problem. Notable examples include the Carlini and Wagner (C&W) loss Carlini and Wagner (2017), the Difference-of-Logits Ratio (DLR) loss Croce and Hein (2020a), and the Minimize the Impact of Floating-point Errors (MIFPE) loss Yu and Xu (2023). The C&W loss, also known as the hinge loss, avoids exponential operations present in CE, reducing the risk of floating-point errors, but it discards some logit elements, potentially weakening the attack's effectiveness. Similarly, the DLR loss scales the difference between the largest logits to improve gradient computation but suffers from similar limitations due to partial logit utilization. In contrast, the MIFPE loss Yu and Xu (2023) identifies relative gradient errors—driven by floating-point underflow and rounding, which are exacerbated by the numerical characteristics of the model's output logits—as the core culprit. By applying a fixed scaling factor of $T = 1$ to the logits in a carefully normalized manner, MIFPE empirically reduces these errors, outperforming CE, C&W, and DLR in efficiency and accuracy across defenses. However, its empirical choice of a fixed $T = 1$ lacks a comprehensive theoretical justification, limiting deeper insights into the error dynamics across diverse attack scenarios.

To bridge this gap, we pioneer the first rigorous theoretical analysis of floating-point errors in CE-based gradient attacks, systematically dissecting relative errors across four distinct scenarios: (i) unsuccessful untargeted attacks, (ii) successful untargeted attacks, (iii) unsuccessful targeted attacks, and (iv) successful targeted attacks. This foundational study uncovers novel patterns in numerical instability and derives the optimal scaling factor $T = t^*$ that minimizes error impact in each scenario. Notably, our analysis reveals that $t^*$ closely approximates 1 in unsuccessful untargeted attacks, providing a theoretical justification for MIFPE's empirical choice and addressing prior optimality gaps.

To validate the correctness of our theoretical derivations, we refine MIFPE by incorporating $T = t^*$ into the Theoretical MIFPE (T-MIFPE) loss function, which further reduces floating-point-induced errors. Extensive experiments on CIFAR-10, CIFAR-100, and ImageNet empirically corroborate the theory through improved robustness evaluation overMIFPE.

In summary, our primary contribution is a pioneering theoretical analysis of floating-point-induced errors in gradient-based attacks, with supporting elements as follows:

- The first comprehensive study of relative errors in gradient computations across four attack scenarios—(i) unsuccessful untargeted, (ii) successful untargeted, (iii) unsuccessful targeted, and (iv) successful targeted—revealing numerical instability patterns across diverse attack contexts.

- Derivation of the optimal scaling factor $T = t^*$, which justifies MIFPE's empirical $T = 1$ (in unsuccessful untargeted attacks).

- Experimental evaluation of the T-MIFPE loss function, incorporating $T = t^*$, on CIFAR-10, CIFAR-100, and ImageNet datasets, where consistent but modest improvements over MIFPE validate the correctness of our theoretical analysis.

## 2 PRELIMINARIES & RELATED WORK

### 2.1 NOTATION AND PRELIMINARIES

To establish a consistent theoretical foundation, we formally introduce the core concepts and notation used throughout this work. Consider a deep neural network (DNN) classifier $f_{\boldsymbol{\theta}} : \mathcal{X} \rightarrow \mathbb{R}^K$ parameterized by $\boldsymbol{\theta}$, which maps an input $\mathbf{x} \in \mathcal{X}$ to output logits $\mathbf{z} \in \mathbb{R}^K$, where $K$ denotes the number of classes. The predicted class is determined as $\arg\max f_{\boldsymbol{\theta}}(\mathbf{x})$, and the true label is denoted by $y$.

Given an input $\hat{\mathbf{x}}$ and its corresponding logits $\mathbf{z} = f_{\boldsymbol{\theta}}(\hat{\mathbf{x}})$, we sort the elements of $\mathbf{z}$ in descending order, denoted by $\mathbf{z}_{\pi_1} \geq \mathbf{z}_{\pi_2} \geq \cdots \geq \mathbf{z}_{\pi_K}$. The logit gap between the largest and second-largest logits is defined as $\Delta = \mathbf{z}_{\pi_1} - \mathbf{z}_{\pi_2}$. In our method, we utilize a detached version of this gap, denoted $\Delta_{\text{detach}}$, which is computed by truncating the gradient flow during backpropagation, ensuring it functions as a constant scaling factor that does not affect gradient computations.

Adversarial attacks seek to craft adversarial examples $\hat{\mathbf{x}}$ that cause the model to misclassify, i.e., $\arg\max f_{\boldsymbol{\theta}}(\hat{\mathbf{x}})_i \neq y$. This is achieved by introducing a perturbation $\delta = \hat{\mathbf{x}} - \mathbf{x}$, constrained by $\|\delta\|_p \leq \epsilon$, where $\epsilon$ is the perturbation magnitude and $p$ specifies the norm (e.g., $\ell_\infty$, $\ell_2$). The adversarial example is generated by solving the optimization problem:

$$\hat{\mathbf{x}} = \mathbf{x} + \arg\max_{\|\delta\|_p \leq \epsilon} L(f_{\boldsymbol{\theta}}(\mathbf{x} + \delta), y), \tag{1}$$

where $L$ is the loss function, typically cross-entropy (CE) for classification tasks.

White-box attacks assume full knowledge of the model's architecture, parameters, and training data, posing the most stringent challenge for defense mechanisms. Gradient-based methods, such as Projected Gradient Descent (PGD) Madry et al. (2018), are widely employed to solve equation 1. PGD iteratively updates the adversarial example via:

$$\hat{\mathbf{x}}_{i+1} = \text{Proj}_{\mathcal{B}_\epsilon(\mathbf{x})} \left( \hat{\mathbf{x}}_i + \alpha_i \cdot \text{sign} \left( \nabla_{\hat{\mathbf{x}}_i} L(f_{\boldsymbol{\theta}}(\hat{\mathbf{x}}_i), y) \right) \right), \tag{2}$$

where $\hat{\mathbf{x}}_0 = \text{Proj}_{\mathcal{B}_\epsilon(\mathbf{x})}(\mathbf{x} + \mathbf{u})$ is the initial adversarial example, with $\mathbf{u} \sim \text{Uniform}[-\epsilon, \epsilon]$ being a random perturbation. $\alpha_i$ is the step size at iteration $i$. $\text{Proj}_{\mathcal{B}_\epsilon(\mathbf{x})}$ denotes the projection operator that ensures the adversarial example remains within the $\epsilon$-ball centered at $\mathbf{x}$ under the $\ell_p$-norm (typically $\ell_\infty$ or $\ell_2$). $\nabla_{\hat{\mathbf{x}}_i} L$ is the gradient of the loss function $L$ with respect to the adversarial example $\hat{\mathbf{x}}_i$.

The effectiveness of PGD relies on the choice of the loss function $L$, a central focus of our investigation.

## 2.2 RELATED WORK

The susceptibility of deep neural networks (DNNs) to adversarial attacks has spurred extensive research into both attack methodologies and defense strategies, with the primary objectives of evaluating model robustness and enhancing resilience against adversarial perturbations. Adversarial attacks are categorized into white-box and black-box settings, where white-box attacks exploit complete knowledge of the model's architecture, parameters, and training data to craft targeted perturbations. Traditional white-box attack methods, such as the Fast Gradient Sign Method (FGSM) Goodfellow et al. (2015), Basic Iterative Method (BIM) Kurakin et al. (2017), Momentum Iterative Method (MIM) Dong et al. (2018), Projected Gradient Descent (PGD) Madry et al. (2018), and Fast Adaptive Boundary Attack (FAB) Croce and Hein (2020b), predominantly target $\ell_\infty$ norm perturbations. Additionally, methods like Carlini and Wagner (C&W) Carlini and Wagner (2017) and DeepFool Moosavi-Dezfooli et al. (2016) are designed for $\ell_2$ norm attacks. These approaches have been widely adopted to assess adversarial robustness; however, extensive empirical evidence has revealed their significant limitation in overestimating model robustness Croce and Hein (2020a).

To address the pervasive issue of overestimating model robustness, researchers have proposed strategies that integrate multiple attack methods, as relying on any single attack method often fails to provide an accurate assessment of robustness. A prominent example is AutoAttack Croce and Hein (2020a), an ensemble-based approach that combines both white-box and black-box attack strategies. AutoAttack has become the de facto standard for benchmarking adversarial robustness due to its comprehensive evaluation capabilities. However, recent studies have demonstrated that superior attack performance can be achieved without integrating multiple attack methods. A notable example is LAFEAT Yu et al. (2021), which leverages latent feature representations to enhance attack efficacy. While these strategies have significantly alleviated the overestimation of model robustness, they often incur substantial computational overhead. This high computational cost limits their applicability to large-scale or real-time scenarios. Moreover, these methods lack a fundamental analysis of the root causes behind the overestimation of robustness in gradient-based attacks. Research by Gao et al. (2022) has shown that these approaches still exhibit significant overestimation issues when evaluating advanced defense strategies, such as ensemble defenses Kariyappa and Qureshi (2019); Pang et al. (2019); Yang et al. (2020).

In an effort to uncover the root causes of robustness overestimation in gradient-based attacks, Yu et al. Yu and Xu (2023) identified that floating-point arithmetic errors introduce relative errors in the computed gradients, which contribute to the overestimation problem. To address this, they proposed a novel loss function, MIFPE (Minimizing Floating-Point Error), designed to mitigate the negative impact of floating-point errors on gradient-based attacks. The MIFPE loss function is defined as:

$$\mathcal{L}^{\text{MIFPE}} (\mathbf{z}, y) \triangleq \mathcal{L}^{\text{CE}} (T \cdot \mathbf{z}/\Delta_{\text{detach}}, y) , \tag{3}$$

While MIFPE empirically demonstrates effectiveness with $T = 1$, this choice lacks theoretical justification, motivating our rigorous analysis of relative gradient errors induced by floating-point arithmetic.

## 3 THEORY ANALYSIS

Adversarial attacks are broadly classified into untargeted and targeted variants, distinguished by their objectives and loss formulations. An untargeted attack employs the cross-entropy loss $CE(\mathbf{z}, y)$ to maximize the deviation of the model's prediction from the true label $y$, inducing misclassification into any incorrect class. In contrast, a targeted attack leverages the negative cross-entropy loss $-CE(\mathbf{z}, y_t)$, where $y_t$ is the attacker-specified target label, aiming to steer the prediction precisely toward $y_t$. This distinction necessitates separate analyses of the relative gradient errors—arising from floating-point arithmetic inaccuracies—when $CE$ serves as the loss function, as the attack type influences the gradient computation.

To comprehensively analyze floating-point-induced relative errors in gradient computations across adversarial attack scenarios, we define four distinct error metrics: (i) $\delta_{u\text{-}u}$ for untargeted attacks in unsuccessful phases, (ii) $\delta_{u\text{-}s}$ for untargeted attacks in successful phases, (iii) $\delta_{t\text{-}u}$ for targeted attacks during unsuccessful attempts, and (iv) $\delta_{t\text{-}s}$ for successful targeted attacks, consistently used throughout our analysis. While robustness evaluation typically terminates optimization upon achieving misclassification (end of the unsuccessful phase), examining both successful and unsuccessful phases is essential to understand numerical instability patterns fully. This holistic approach reveals how errors evolve as logits transition from correct to incorrect classifications and uncovers post-misclassification gradient vulnerabilities.

### 3.1 RELATIVE ERROR OF GRADIENT IN UNTARGETED ATTACKS

First, we examine the relative error in the computed gradients due to floating-point inaccuracies under untargeted attacks. In untargeted adversarial attacks, the attacker's primary goal is to maximize the value of $\max_{i \neq y} \mathbf{z}_i - \mathbf{z}_y$, transforming it from a negative value (indicating correct classification) to a positive value (indicating misclassification).

#### 3.1.1 UNSUCCESSFUL ATTACK PHASE

When $\mathbf{z}_y = \mathbf{z}_{\pi_1}$

$$\max_{i \neq y} \mathbf{z}_i - \mathbf{z}_y = \mathbf{z}_{\pi_2} - \mathbf{z}_{\pi_1} \tag{4}$$

$$CE (\mathbf{z}, y) = - \log p_y = - \log \frac{e^{\mathbf{z}_y - \mathbf{z}_{\pi 1}}}{\sum_{i=1}^{K} e^{\mathbf{z}_i - \mathbf{z}_{\pi 1}}} \tag{5}$$

$$CE (c\mathbf{z}, y) = - \log p_y^c = - \log \frac{e^{c(\mathbf{z}_y - \mathbf{z}_{\pi_1})}}{\sum_{i=1}^{K} e^{c(\mathbf{z}_i - \mathbf{z}_{\pi_1})}} \tag{6}$$

where $c = \frac{t}{\Delta_{\text{detach}}}$ is a scale factor, $t > 0$, and $p_i^c = e^{c(\mathbf{z}_i - \mathbf{z}_{\pi 1})} / \sum_{j=1}^{K} e^{c(\mathbf{z}_j - \mathbf{z}_{\pi 1})}$.

$$\nabla_{\hat{\mathbf{z}}} CE(\mathbf{z}, y) = c \left( -1 + p_y^c \right) \nabla_{\hat{\mathbf{x}}} \left( \mathbf{z}_y - \mathbf{z}_{\pi_1} \right) + \sum_{i \neq y} c p_i^c \nabla_{\hat{\mathbf{x}}} \left( \mathbf{z}_i - \mathbf{z}_{\pi_1} \right)$$

$$= c \sum_{i \neq \pi_1} p_i^c \nabla_{\hat{\mathbf{x}}} \left( \mathbf{z}_i - \mathbf{z}_{\pi_1} \right)$$

$$= c p_{\pi_2}^c \nabla_{\hat{\mathbf{x}}} \left( \mathbf{z}_{\pi_2} - \mathbf{z}_{\pi_1} \right) + c p_{\pi_3}^c \nabla_{\hat{\mathbf{x}}} \left( \mathbf{z}_{\pi_3} - \mathbf{z}_{\pi_2} + \mathbf{z}_{\pi_2} - \mathbf{z}_{\pi_1} \right)$$

$$+ \cdots + c p_{\pi_K}^c \nabla_{\hat{\mathbf{x}}} \left( \mathbf{z}_{\pi_K} - \mathbf{z}_{\pi_{K-1}} + \cdots + \mathbf{z}_{\pi_2} - \mathbf{z}_{\pi_1} \right)$$

$$= c(1 - p_{\pi_1}^c)\nabla_{\hat{\mathbf{x}}}\left(\mathbf{z}_{\pi_2} - \mathbf{z}_{\pi_1}\right) + c(1 - p_{\pi_1}^c - p_{\pi_2}^c)\nabla_{\hat{\mathbf{x}}}\left(\mathbf{z}_{\pi_3} - \mathbf{z}_{\pi_2}\right)$$
$$+ \cdots + c(1 - p_{\pi_1}^c - \cdots - p_{\pi_{K-1}}^c)\nabla_{\hat{\mathbf{x}}}\left(\mathbf{z}_{\pi_K} - \mathbf{z}_{\pi_{K-1}}\right) \tag{7}$$

As derived from the primary objective of untargeted attacks in Equation equation 4, the gradient term $\nabla_{\hat{\mathbf{x}}}(\mathbf{z}_{\pi_2} - \mathbf{z}_{\pi_1})$ emerges as a critical component. During the unsuccessful attack phase, we consequently focus on minimizing the relative error in $|c(1 - p_{\pi_1}^c)\nabla_{\hat{\mathbf{x}}}(\mathbf{z}_{\pi_2} - \mathbf{z}_{\pi_1})|$.

To formally analyze the impact of floating-point errors, we define the relative error $\delta(t)_{u\_u}$ in the computed gradient magnitude. Let $r = |c(1 - p_{\pi_1}^c)\nabla_{\hat{\mathbf{x}}}(\mathbf{z}_{\pi_2} - \mathbf{z}_{\pi_1})|$ denote the exact value of the gradient magnitude in real arithmetic. Its floating-point approximation is given by $\mathrm{fl}(r)$, and the absolute floating-point error is defined as:

$$\epsilon = |r - \mathrm{fl}(r)|. \tag{8}$$

The relative error in the gradient computation is then expressed as:

$$\delta(t)_{u\_u} = \frac{\epsilon}{|c(1 - p_{\pi_1}^c)\nabla_{\hat{\mathbf{x}}}(\mathbf{z}_{\pi_2} - \mathbf{z}_{\pi_1})|} = \frac{\epsilon}{r}. \tag{9}$$

We analyze three critical scenarios based on the value of $r$:

**Case 1: Underflow.** When $r$ is smaller than the smallest representable positive value of the floating-point format, underflow occurs, resulting in $\mathrm{fl}(r) = 0$. Consequently, the absolute error equals the exact value ($\epsilon = r$), and the relative error reaches its maximum: $\delta(t)_{u\_u} = 1$. **Case 2: Overflow.** If $r$ exceeds the largest representable finite value, overflow occurs, producing $\mathrm{fl}(r) = \mathrm{NaN}$. This renders the gradient computation invalid and the attack process unstable. Since this scenario prevents meaningful robustness evaluation, we exclude it from our theoretical analysis. **Case 3: Normal Range with Truncation Error.** When $r$ lies within the normal range of the floating-point format, the error $\epsilon$ arises from numerical truncation during computation. While the exact value of $\epsilon$ depends on implementation-specific rounding, it is bounded by a deterministic maximum $\epsilon_{\max}$, defined by the floating-point precision (e.g., $\epsilon_{\max} = 2^{-23}$ for 32-bit float). To ensure a rigorous and reproducible worst-case analysis, we focus on the supremum of the relative error:

$$\delta(t)_{u\_u}^{\sup} = \sup_{\epsilon \in [0, \epsilon_{\max}]} \delta(t, \epsilon)_{u\_u} = \frac{\epsilon_{\max}}{r}. \tag{10}$$

This approach allows us to derive a robust upper bound for the relative error, establishing a reliable framework for analyzing the impact of floating-point arithmetic on adversarial attacks.

$$\delta(t)_{u\_u}^{\sup} = \sup_{\epsilon \in [0, \epsilon_{\max}]} \delta(t, \epsilon)_{u\_u} = \frac{\epsilon_{\max}}{|c(1 - p_y^c)\nabla_{\hat{\mathbf{x}}}(\mathbf{z}_{\pi_2} - \mathbf{z}_{\pi_1})|} \tag{11}$$

Consequently, the minimum value of $\delta(t)_{u\_u}^{\sup}$ can be expressed as:

$$\delta(t)_{u\_u}^{\sup\_\min} = \frac{\epsilon_{\max}}{|c(1 - p_{\pi_1}^c)\nabla_{\hat{\mathbf{x}}}(\mathbf{z}_{\pi_2} - \mathbf{z}_{\pi_1})|_{max}}, \tag{12}$$

where $|c(1 - p_{\pi_1}^c)\nabla_{\hat{\mathbf{x}}}(\max_{i \neq \pi_1}\mathbf{z}_i - \mathbf{z}_{\pi_1})|_{max}$ denotes the maximum value of the denominator across the relevant domain.

Gradient-based iterative attacks involve repeatedly applying a uniform procedure at each iteration, where perturbations are introduced to the input data based on gradient information derived through backpropagation. Given the repetitive nature of this mechanism, the overall multi-iteration process can be effectively understood by analyzing a single iteration in detail. Consequently, we focus our analysis on the relative error incurred during the gradient computation phase via backpropagation within a specific iteration of the multi-iteration attack. Notably, the gradient $\nabla_{\hat{\mathbf{x}}}(\mathbf{z}_{\pi_2} - \mathbf{z}_{\pi_1})$, computed in this process, depends solely on the model's internal parameters and remains invariant to the scaling factor $t/\Delta_{\text{detach}}$ incorporated into the loss function. Thus, we treat $\nabla_{\hat{\mathbf{x}}}(\mathbf{z}_{\pi_2} - \mathbf{z}_{\pi_1})$ as a constant with respect to $t/\Delta_{\text{detach}}$. As a result, maximizing $|c(1 - p_{\pi_1}^c)\nabla_{\hat{\mathbf{x}}}(\mathbf{z}_{\pi_2} - \mathbf{z}_{\pi_1})|$ simplifies to maximizing $c(1 - p_{\pi_1}^c)$, since the gradient term is constant. To analyze the maximum value of $c(1 - p_{\pi_1}^c)$, we define a new function $g(t)_{u\_u}$ as follows:

$$g(t)_{u\_u} = c\left(1 - p_{\pi_1}^c\right) = c\left(1 - \frac{1}{B}\right) > 0 \tag{13}$$

where $p_{\pi_1}^c = \frac{e^{c \cdot 0}}{\sum_{j=1}^{K} e^{c(\mathbf{z}_j - \mathbf{z}_{\pi_1})}} = \frac{1}{B}$, $B = 1 + \sum_{j \neq \pi_1}^{K} e^{c(\mathbf{z}_j - \mathbf{z}_{\pi_1})} > 1$, $c = t/\Delta_{\text{detach}}$, and $\Delta_{\text{detach}} = \mathbf{z}_{\pi_1} - \mathbf{z}_{\pi_2} > 0$.

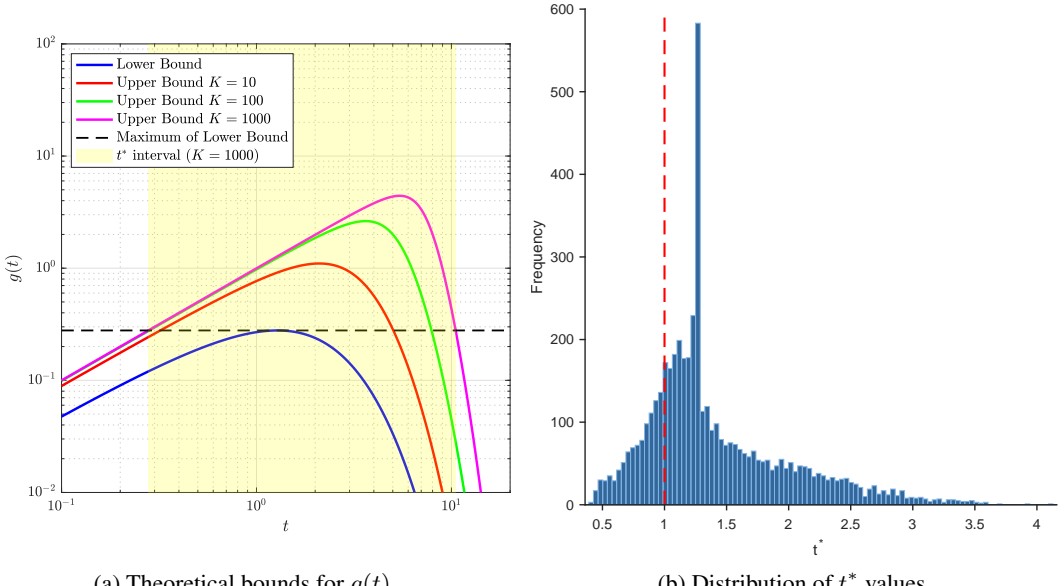

(a) Theoretical bounds for $g(t)_{u\_u}$.

(b) Distribution of $t^*$ values.

Figure 1: Bounds analysis and optimal scaling factor distribution for unsuccessful untargeted attacks. (a) Logarithmic plot of the lower bound $\frac{te^{-t}}{1+e^{-t}}$ and upper bounds $\frac{(K-1)te^{-t}}{1+(K-1)e^{-t}}$ for $g(t)_{u\_u}$ across $K = \{10, 100, 1000\}$, with $t \in [0.1, 20]$ and $g(t)_{u\_u} \in [10^{-2}, 10^2]$. The dashed line marks the maximum of the lower bound. The shaded region denotes the interval $[0.279, 10.512]$ where $t^*$ maximizes $g(t)_{u\_u}$ for $K = 1000$. (b) Distribution of $t^*$ values (averaged over 100 bins) for defending models Engstrom et al. (2019) on ImageNet.

Under the condition that $K \geq 2$, the term $B$ satisfies $1 + e^{-t} \leq B < 1 + (K-1)e^{-t}$. This follows from the fact that there are $K - 1$ terms in the sum, each bounded above by $e^{-t}$ (since $c(\mathbf{z}_j - \mathbf{z}_{\pi_1}) \leq -t$ for $j \neq \pi_1$), yielding the upper bound on $B$. The lower bound assumes at least one term (corresponding to the second-largest logit) achieves approximately $e^{-t}$, with others negligible. Consequently, the lower bound for $g(t)_{u\_u}$ is $\frac{te^{-t}}{1+e^{-t}}$, and the upper bound is $\frac{(K-1)te^{-t}}{1+(K-1)e^{-t}}$, as shown in Figure 1a.

The maximum value of $g(t)_{u\_u}$ lies within the interval where the upper bound exceeds the maximum of the lower bound. The lower bound achieves its maximum at $t \approx 1.278$. Solving the upper bound equal to $k$ gives the interval endpoints. When $K = 10$, it approximates $(0.321, 5.035)$. when $K = 100$, it approximates $(0.282, 7.905)$. when $K = 1000$, it approximates $(0.279, 10.512)$.

To determine the value of $t^*$ that maximizes $g(t)_{u\_u}$ for classification tasks with $K \leq 1000$, a grid search is performed over the interval $(0.279, 10.512)$, discretized into 1000 equally spaced points. Leveraging GPU-accelerated parallel computation, the optimal $t^*$ is efficiently identified, with negligible overhead (0.044473 seconds for 10,000 samples in a single attack iteration). As shown in Figure 1b, the derived $t^*$ values predominantly cluster around $t \approx 1.278$, closely approximating MIFPE's empirical choice of $T = 1$. This near-optimality theoretically justifies MIFPE's strong performance, as $T = 1$ is already near-optimal for maximizing $g(t)_{u\_u}$. To enhance stability and mitigate frequent fluctuations in $t^*$ while preserving theoretical optimality, we apply $t^* = \max\{1.278, t^*\}$, where 1.278 represents the maximum point of the lower bound function.

### 3.1.2 SUCCESSFUL ATTACK PHASE

Due to space constraints, the detailed analysis of the relative gradient error $\delta_{u\text{-}s}$ for successful untargeted attacks is deferred to Appendix 6.1.

## 3.2 RELATIVE ERROR OF GRADIENT IN TARGETED ATTACKS

We now examine the relative error in gradients induced by floating-point inaccuracies during targeted attacks, where the objective is to maximize $\mathbf{z}_{y_t} - \max_{i \neq y_t} \mathbf{z}_i$, shifting it from negative (unsuccessful) to positive (successful) values. Detailed derivations of $\delta_{t\text{-}u}$ for unsuccessful targeted attacks and $\delta_{t\text{-}s}$ for successful ones are provided in Appendices 6.2 and 6.3, respectively.

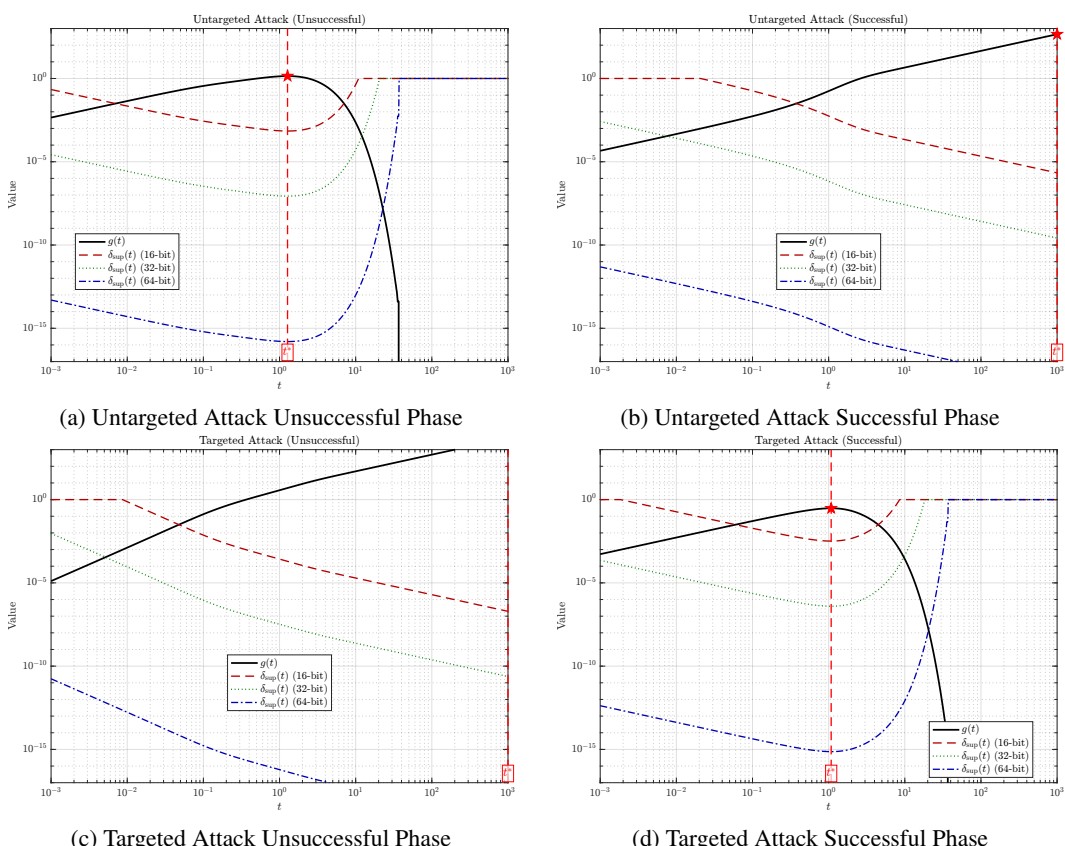

(a) Untargeted Attack Unsuccessful Phase

(b) Untargeted Attack Successful Phase

(c) Targeted Attack Unsuccessful Phase

(d) Targeted Attack Successful Phase

Figure 2: Analysis of the relative error in gradients computed using cross-entropy loss for a 10-class classifier under targeted and untargeted attack scenarios. **Untargeted Attack Scenario** : (a) *Untargeted Attack Unsuccessful Attack Phase* ($\mathbf{z}_y = \mathbf{z}_{\pi_1}$): ($g(t) = c(1 - p^c_{\pi_1})$) Logits $\mathbf{z} = [2.5, -1.3, 0.8, 3.8, -0.9, 1.7, -2.1, 3.6, 0.4, -1.5]$, with $y = 3$ as the ground-truth label and $\mathbf{z}_{\pi_1} = 3.8$. (b) *Untargeted Attack Successful Attack Phase* ($\mathbf{z}_y \neq \mathbf{z}_{\pi_1}$): ($g(t) = cp^c_{\pi_1}$) Logits $\mathbf{z} = [1.2, -1.5, 0.5, 1.9, -1.2, 4.2, -2.3, 2.0, 0.1, -1.8]$, with $y = 5$ as the misclassification label and $\mathbf{z}_{\pi_1} = 4.2$. **Targeted Attack Scenario** : (c) *Targeted Attack Unsuccessful Attack Phase* ($\mathbf{z}_{y_t} \neq \mathbf{z}_{\pi_1}$): ($g(t) = c(p^c_{\pi_1} - p^c_{y_t})$)Original logits $\mathbf{z} = [2.5, -1.3, 0.8, 3.8, -0.9, 1.7, -2.1, 3.6, 0.4, -1.5]$, with $y = 3$ (correct) and target label $y_t \neq 3$, where $\mathbf{z}_{\pi_1} = 3.8$. (d) *Targeted Attack Successful Attack Phase* ($\mathbf{z}_{y_t} = \mathbf{z}_{\pi_1}$): ($g(t) = c(1 - p^c_{\pi_1})$) Perturbed logits $\mathbf{z} = [1.0, 4.5, 0.3, 1.2, -1.0, 1.5, -2.5, 2.8, 0.0, -1.8]$, achieving targeted misclassification to $y_t = 1$, with $\mathbf{z}_{\pi_1} = 4.5$. Each subplot illustrates: (1) Upper bounds of the relative error $\delta_{\text{sup}}(t)$ for 16-bit (red dashed), 32-bit (blue dotted), and 64-bit (green dash-dotted) floating-point precision; (2) Critical points $t^*$ (red stars) marking the maxima of $g(t)$, where $\delta_{\text{sup}}(t)$ reaches local minima (gray vertical dashed lines).

We visualize the numerical behavior of $g(t)$ and the relative gradient error $\delta_{\sup}(t)$ across four distinct scenarios—(1) untargeted attacks in unsuccessful phases, (2) untargeted attacks in successful phases, (3) targeted attacks in unsuccessful phases, and (4) targeted attacks in successful phases—as depicted in Figure 2. Our theoretical analysis, illustrated in Figure 2, demonstrates that the optimal scaling factor $t^*$, which minimizes floating-point-induced gradient errors, varies with the model's output logits $\mathbf{z}$ and the specific attack scenario. In multi-round gradient-based attacks, iterative input perturbations cause $\mathbf{z}$ to evolve, dynamically shifting $t^*$ across scenarios. Notably, as shown in Figure 2a for unsuccessful untargeted attacks, MIFPE's empirical $T = 1$ closely approximates the theoretically derived $t^*$, substantially reducing relative gradient errors. Similarly, in unsuccessful targeted attacks (Figure 2c), the relative error decreases as $T$ increases, with $T = 1$ significantly reducing error compared to $T < 1$. These findings suggest that MIFPE's $T = 1$ is near-optimal in key scenarios, limiting the scope for substantial further improvements when using $t^*$. This theoretical insight underpins our subsequent experimental validation, where modest gains from substituting $t^*$ for $T = 1$ confirm the precision of our analysis.

To further validate the correctness of our theoretical analysis—which derives $t^*$ as the value minimizing relative errors—we replace MIFPE's fixed $T = 1$ with the scenario-adaptive $t^*$ (recomputed before each iteration based on updated $\mathbf{z}$). Any improvements achieved thereby would empirically corroborate our theoretical derivations. Leveraging this validation approach, we introduce the Theoretical MIFPE, denoted as T-MIFPE, defined as follows:

$$\mathcal{L}^{\text{T-MIFPE}}\left(\mathbf{z}, y\right) \triangleq \mathcal{L}^{\text{ce}}\left(\frac{t^*\mathbf{z}}{(\mathbf{z}_{\pi_1} - \mathbf{z}_{\pi_2})_{\text{detach}}}, y\right) \tag{14}$$

$$\mathcal{L}_{\text{target}}^{\text{T-MIFPE}}\left(\mathbf{z}, y_{\text{t}}\right) \triangleq -\mathcal{L}^{\text{ce}}\left(\frac{t^*\mathbf{z}}{(\mathbf{z}_{\pi_1} - \mathbf{z}_{\pi_2})_{\text{detach}}}, y_{\text{t}}\right) \tag{15}$$

## 4 EXPERIMENTS

To validate the correctness of our theoretical analysis—which derives the optimal scaling factor $t^*$ to minimize floating-point-induced relative gradient errors in CE-based attacks—we evaluate the T-MIFPE loss function, incorporating $t^*$ to refine MIFPE's empirical $T = 1$. Our experiments compare T-MIFPE against the Cross-Entropy (CE) baseline—selected due to our analysis targeting CE's floating-point error issues—and MIFPE as a direct benchmark for our theoretical refinements. Experiments employ $\ell_\infty$-bounded PGD attacks on CIFAR-10, CIFAR-100 Krizhevsky et al. (2010), and ImageNet Deng et al. (2009) datasets, with a fixed random seed of 0 for reproducibility. We evaluate both untargeted attacks (100 iterations) and multi-targeted attacks (targeting the 9 closest incorrect classes, with 100 iterations per target, totaling 900 iterations), using a cosine step-size schedule $\epsilon_i = \epsilon(1 + \cos(\pi i/I))$ ($I = 100$) and momentum of 0.75 Croce and Hein (2020a), retaining the strongest adversarial examples. To demonstrate that mitigating floating-point errors reduces robustness overestimation, we compare T-MIFPE's robust accuracy to that of RobustBench's AutoAttack Croce et al. (2020) (using 4900 iterations), the standard for evaluating model robustness, highlighting the minimal gap achieved by our theoretically grounded approach.

Results for untargeted and multi-targeted attacks are presented in Table 1 and Table 2, respectively. Across all datasets, T-MIFPE consistently outperforms CE, addressing its floating-point-induced gradient errors, and yields consistent improvements over MIFPE, further validating our theoretical derivation of $t^*$. For untargeted attacks, T-MIFPE achieves 25.38% on ImageNet Wong et al. (2020) versus MIFPE's 25.71% (0.34% improvement), 24.96% on CIFAR-100 Sitawarin et al. (2020) versus 25.24% (0.28% improvement), and 55.09% on CIFAR-10 Hendrycks et al. (2019) versus 55.12% (0.03% improvement). For multi-targeted attacks, T-MIFPE achieves 25.02% on ImageNet Wong et al. (2020) versus 25.06% (0.04% improvement), 18.92% on CIFAR-100 Rice et al. (2020) versus 18.95% (0.03% improvement), and 53.30% on CIFAR-10 Huang et al. (2020) versus 53.31% (0.01% improvement). As $t^* \approx 1$ in unsuccessful untargeted attacks (Figure 1b) and targeted attacks (Figure 2c) implies the relative error decreases as $T$ increases, with $T = 1$ significantly reducing error compared to $T < 1$, so MIFPE's $T = 1$ is near-optimal, limiting the scope for substantial further improvements, these consistent gains robustly corroborate our theoretical advancements in reducing floating-point-induced gradient errors.

Table 1: Comparing the proposed T-MIFPE loss ($\mathcal{L}^{\text{T-MIFPE}}$), against CE ($\mathcal{L}^{\text{ce}}$), and MIFPE($\mathcal{L}^{\text{MIFPE}}$) losses under untargeted attacks. For each surrogate loss, we use PGD-100 with the step-size schedule $\epsilon_i = \epsilon(1 + cos(\pi i/I))$, where $I = 100$ denotes the total iterations and $i$ represents the current iteration. and momentum $\nu = 0.75$ . Numbers in parentheses indicate the improvement w.r.t the CE baseline. The AutoAttack, calculated using an ensemble of attacks and a minimum of 4900 iterations, is reported from RobustBench to demonstrate how closely T-MIFPE can approach the lowest known robustness accuracy with only 100 iterations.

| Defense method | Architecture | Clean | APGD-CE 100 | CE ($\mathcal{L}^{\text{ce}}$) 100 | MIFPE ($\mathcal{L}^{\text{MIFPE}}$) 100 | | T-MIFPE ($\mathcal{L}^{\text{T-MIFPE}}$) 100 | | AutoAttack 4900 |
|---|---|---|---|---|---|---|---|---|---|
| **CIFAR-10**, $\ell_\infty$, $\varepsilon = 8/255$ | | | | | | | | | |
| Uncovering limits Gowal et al. (2020) | WRN-70-16 | 91.10 | 67.96 | 67.96 | 65.96 | (-2.00) | **65.95** | $(-2.01)$ | 65.87 |
| Fixing data augmentation Rebuffi et al. (2021) | WRN-106-16 | 88.50 | 67.48 | 67.57 | 64.78 | (-2.79) | **64.72** | $(-2.85)$ | 64.58 |
| Fixing data augmentation Rebuffi et al. (2021) | WRN-70-16 | 88.54 | 67.14 | 67.27 | 64.57 | (-2.80) | **64.46** | $(-2.81)$ | 64.20 |
| Uncovering limits Gowal et al. (2020) | WRN-28-10 | 89.48 | 65.63 | 65.59 | 62.97 | (-2.62) | **62.94** | $(-2.65)$ | 62.76 |
| Adversarial weight perturbation Wu et al. (2021) | WRN-28-10 | 88.25 | 63.20 | 63.18 | 60.10 | (-3.08) | **60.09** | $(-3.09)$ | 60.04 |
| Unlabeled data Carmon et al. (2019) | WRN-28-10 | 89.69 | 61.87 | 61.60 | 59.73 | (-1.87) | **59.70** | $(-1.90)$ | 59.53 |
| HYDRA Sehwag et al. (2020) | WRN-28-10 | 88.98 | 59.68 | 59.53 | 57.39 | (-2.14) | **57.36** | $(-2.17)$ | 57.14 |
| Pre-training Hendrycks et al. (2019) | WRN-28-10 | 87.11 | 57.10 | 57.07 | 55.12 | (-1.95) | **55.09** | $(-1.98)$ | 54.92 |
| Overfitting Rice et al. (2020) | WRN-34-20 | 85.34 | 56.74 | 56.85 | 53.67 | (-3.18) | **53.66** | $(-3.19)$ | 53.42 |
| Self-adaptive training Huang et al. (2020)‡ | WRN-34-10 | 83.48 | 56.22 | 56.12 | 53.652 | (-2.60) | **53.51** | $(-2.61)$ | 53.34 |
| **CIFAR-100**, $\ell_\infty$, $\epsilon = 8/255$ | | | | | | | | | |
| Adversarial weight perturbation Wu et al. (2020b) | WRN-34-10 | 60.38 | 33.15 | 33.09 | 29.32 | (-3.77) | **29.22** | $(-3.87)$ | 28.86 |
| Pre-training Hendrycks et al. (2019) | WRN-28-10 | 59.23 | 32.22 | 32.82 | 29.10 | (-3.72) | **28.96** | $(-4.14)$ | 28.42 |
| Progressive Hardening Sitawarin et al. (2020) | WRN-34-10 | 62.82 | 26.60 | 26.18 | 25.24 | ( 0.94) | **24.96** | $(-1.22)$ | 24.57 |
| Overfitting Rice et al. (2020) | RN-18 | 53.83 | 20.72 | 20.47 | 19.40 | (-1.07) | **19.27** | $(-1.20)$ | 18.95 |
| **ImageNet**, $\ell_\infty$, $\epsilon = 4/255$ | | | | | | | | | |
| Transfer Better Salman et al. (2020) | RN-50 | 64.02 | 38.42 | 38.44 | 35.16 | (-3.28) | **34.90** | $(-3.54)$ | 34.96 |
| Robustness library Engstrom et al. (2019) | RN-50 | 62.56 | 32.42 | 32.16 | 30.08 | (-2.08) | **29.68** | $(-2.48)$ | 29.22 |
| Fast adversarial training Wong et al. (2020) | RN-50 | 53.30 | 27.26 | 27.12 | 25.71 | (-1.41) | **25.38** | $(-1.74)$ | 25.24 |
| Transfer Better Salman et al. (2020) | RN-18 | 52.92 | 29.28 | 29.30 | 25.60 | ( -3.66) | **25.48** | $(-3.82)$ | 25.26 |

Furthermore, T-MIFPE's performance demonstrates reduced robustness overestimation compared to established benchmarks. For untargeted attacks with 100 iterations, T-MIFPE closely approximates AutoAttack's results. For instance, on ImageNet with model Wong et al. (2020), T-MIFPE achieves 25.38% robust accuracy versus AutoAttack's 25.24% (0.14% gap); on CIFAR-100 with model Sitawarin et al. (2020), 24.96% versus 25.57% (0.39% gap); and on CIFAR-10 with model Hendrycks et al. (2019), 55.09% versus 54.92% (0.17% gap). For multi-targeted attacks (900 iterations), T-MIFPE often surpasses AutoAttack's performance, e.g., on ImageNet Wong et al. (2020), 25.02% versus 25.24% (0.22% improvement); on CIFAR-100 Rice et al. (2020), 18.92% versus 18.95% (0.03% improvement); and on CIFAR-10 Huang et al. (2020), 53.30% versus 53.34% (0.04% improvement)—especially T-MIFPE's ability to match or exceed AutoAttack's performance with fewer iterations—further validating the correctness of our theoretical analysis.

## 5 CONCLUSION

This work establishes a pioneering theoretical framework that systematically analyzes floating-point-induced relative errors in gradient computations for CE-based adversarial attacks, focusing on four distinct scenarios: (i) unsuccessful untargeted attacks, (ii) successful untargeted attacks, (iii) unsuccessful targeted attacks, and (iv) successful targeted attacks. Our comprehensive analysis uncovers novel patterns of numerical instability and derives the optimal scaling factor $t^*$, providing a rigorous foundation for understanding and mitigating gradient errors. Notably, our finding that $t^* \approx 1$ in unsuccessful untargeted attacks validates the empirical efficacy of MIFPE's $T = 1$, addressing prior optimality gaps. To empirically confirm these theoretical insights, we propose the Theoretical Minimize the Impact of Floating Point Error (T-MIFPE) loss function, incorporating $t^*$ to refine MIFPE. Experiments on CIFAR-10, CIFAR-100, and ImageNet datasets demonstrate that T-MIFPE reduces robustness overestimation, achieving performance comparable to AutoAttack while requiring significantly fewer iterations. These results, though modest in improvement due to the near-optimality of $T = 1$, robustly corroborate the precision of our theoretical derivations, offering a generalizable approach for designing numerically stable loss functions and advancing reliable adversarial robustness evaluation.

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

# 6 TECHNICAL APPENDICES AND SUPPLEMENTARY MATERIAL

## 6.1 SUCCESSFUL ATTACK PHASE OF UNTARGETED ATTACKS

When $\mathbf{z}_y \neq \mathbf{z}_{\pi_1}$, Here, we assume $\mathbf{z}_y = \mathbf{z}_{\pi_j}$ for some $j \in \{2, \ldots, K\}$.

$$CE(\mathbf{z}, y) = -\log p_y = -\log \frac{e^{\mathbf{z}_y - \mathbf{z}_{\pi_2}}}{\sum_{i=1}^{K} e^{\mathbf{z}_i - \mathbf{z}_{\pi_2}}} \tag{16}$$

$$CE(c\mathbf{z}, y) = -\log p_y^c = -\log \frac{e^{c(\mathbf{z}_y - \mathbf{z}_{\pi_2})}}{\sum_{i=1}^{K} e^{c(\mathbf{z}_i - \mathbf{z}_{\pi_2})}} \tag{17}$$

$$
\begin{aligned}
\nabla_{\hat{\mathbf{x}}} CE(c\mathbf{z}, y) &= c\left(-1 + p_y^c\right) \nabla_{\hat{\mathbf{x}}} (\mathbf{z}_y - \mathbf{z}_{\pi_2}) + \sum_{i \neq y} c p_i^c \nabla_{\hat{\mathbf{x}}} (\mathbf{z}_i - \mathbf{z}_{\pi_2}) \\
&= c\left(-1 + p_{\pi_j}^c\right) \nabla_{\hat{\mathbf{x}}} (\mathbf{z}_{\pi_j} - \mathbf{z}_{\pi_2}) + \sum_{i \neq j} c p_{\pi_i}^c \nabla_{\hat{\mathbf{x}}} (\mathbf{z}_{\pi_i} - \mathbf{z}_{\pi_2}) \\
&= c\left(-1 + p_{\pi_j}^c\right) \nabla_{\hat{\mathbf{x}}} (\mathbf{z}_{\pi_j} - \mathbf{z}_{\pi_{j-1}} + \ldots + \mathbf{z}_{\pi_3} - \mathbf{z}_{\pi_2}) \\
&\quad + c p_{\pi_1}^c \nabla_{\hat{\mathbf{x}}} (\mathbf{z}_{\pi_1} - \mathbf{z}_{\pi_2}) \\
&\quad + c p_{\pi_3}^c \nabla_{\hat{\mathbf{x}}} (\mathbf{z}_{\pi_3} - \mathbf{z}_{\pi_2}) \\
&\quad + \ldots \\
&\quad + c p_{\pi_K}^c \nabla_{\hat{\mathbf{x}}} (\mathbf{z}_{\pi_K} - \mathbf{z}_{\pi_{K-1}} + \ldots + \mathbf{z}_{\pi_3} - \mathbf{z}_{\pi_2}) \\
&= c p_{\pi_1}^c \nabla_{\hat{\mathbf{x}}} (\mathbf{z}_{\pi_1} - \mathbf{z}_{\pi_2}) \\
&\quad + c(1 - p_{\pi_j}^c - p_{\pi_3}^c - \ldots - p_{\pi_K}^c) \nabla_{\hat{\mathbf{x}}} (\mathbf{z}_{\pi_2} - \mathbf{z}_{\pi_3}) \\
&\quad + \ldots \\
&\quad + c(1 - p_{\pi_j}^c - p_{\pi_j}^c - \ldots - p_{\pi_K}^c) \nabla_{\hat{\mathbf{x}}} (\mathbf{z}_{\pi_{j-1}} - \mathbf{z}_{\pi_j}) \\
&\quad + c(p_{\pi_{j+1}}^c + \ldots + p_{\pi_K}^c) \nabla_{\hat{\mathbf{x}}} (\mathbf{z}_{\pi_{j+1}} - \mathbf{z}_{\pi_j}) \\
&\quad + \ldots \\
&\quad + c p_{\pi_K}^c \nabla_{\hat{\mathbf{x}}} (\mathbf{z}_{\pi_K} - \mathbf{z}_{\pi_{K-1}}) \\
&= c p_{\pi_1}^c \nabla_{\hat{\mathbf{x}}} (\mathbf{z}_{\pi_1} - \mathbf{z}_{\pi_2}) \\
&\quad + c(p_{\pi_1}^c + p_{\pi_2}^c - p_{\pi_j}^c) \nabla_{\hat{\mathbf{x}}} (\mathbf{z}_{\pi_2} - \mathbf{z}_{\pi_3}) \\
&\quad + \ldots \\
&\quad + c(p_{\pi_1}^c + \ldots + p_{\pi_{j-1}}^c - p_{\pi_j}^c) \nabla_{\hat{\mathbf{x}}} (\mathbf{z}_{\pi_{j-1}} - \mathbf{z}_{\pi_j}) \\
&\quad + c(p_{\pi_{j+1}}^c + \ldots + p_{\pi_K}^c) \nabla_{\hat{\mathbf{x}}} (\mathbf{z}_{\pi_{j+1}} - \mathbf{z}_{\pi_j}) \\
&\quad + \ldots \\
&\quad + c p_{\pi_K}^c \nabla_{\hat{\mathbf{x}}} (\mathbf{z}_{\pi_K} - \mathbf{z}_{\pi_{K-1}})
\end{aligned} \tag{18}
$$

where $c = \frac{t}{\Delta_{\text{detach}}}$ is a scale factor, $t > 0$, and $p_{\pi_i}^c = e^{c(\mathbf{z}_{\pi_i} - \mathbf{z}_{\pi_2})} / \sum_{j=1}^{K} e^{c(\mathbf{z}_{\pi_j} - \mathbf{z}_{\pi_2})}$.

$$\max_{i \neq y} \mathbf{z}_i - \mathbf{z}_y = \mathbf{z}_{\pi_1} - \mathbf{z}_{\pi_j} = (\mathbf{z}_{\pi_1} - \mathbf{z}_{\pi_2}) + \ldots + (\mathbf{z}_{\pi_{j-1}} - \mathbf{z}_{\pi_j}) \tag{19}$$

Based on the primary objective of the untargeted attack in the Successful Attack Phase, as defined in Equation equation 19, the gradients $\nabla_{\hat{\mathbf{x}}} (\mathbf{z}_{\pi_1} - \mathbf{z}_{\pi_2})$, $\nabla_{\hat{\mathbf{x}}} (\mathbf{z}_{\pi_2} - \mathbf{z}_{\pi_3})$, ..., $\nabla_{\hat{\mathbf{x}}} (\mathbf{z}_{\pi_{j-1}} - \mathbf{z}_{\pi_j})$ are critical components. To enhance the accuracy of gradient computations in such attacks, it is necessary to simultaneously minimize the upper bounds of the relative errors associated with the terms $|c p_{\pi_1}^c \nabla_{\hat{\mathbf{x}}} (\mathbf{z}_{\pi_1} - \mathbf{z}_{\pi_2})|$, $|c(p_{\pi_1}^c + p_{\pi_2}^c - p_{\pi_j}^c) \nabla_{\hat{\mathbf{x}}} (\mathbf{z}_{\pi_2} - \mathbf{z}_{\pi_3})|$, ..., $|c(p_{\pi_1}^c + \cdots + p_{\pi_{j-1}}^c - p_{\pi_j}^c) \nabla_{\hat{\mathbf{x}}} (\mathbf{z}_{\pi_{j-1}} -$

$\mathbf{z}_{\pi_j})|$. We define $\delta(t)^{\text{sup\_min}}_{u\_s\_i-1\_i}$ as the upper bound of the relative error in $\nabla_{\hat{\mathbf{x}}}(\mathbf{z}_{\pi_{i-1}} - \mathbf{z}_{\pi_i})$ during the Successful Attack Phase of an untargeted attack, where $i$ ranges from 2 to $j$, and $j \in \{2, ..., K\}$:

When $i = 2$:

$$\delta(t)^{\text{sup\_min}}_{u\_s\_1\_2} = \frac{\epsilon_{\max}}{|cp^c_{\pi_1} \nabla_{\hat{\mathbf{x}}}(\mathbf{z}_{\pi_1} - \mathbf{z}_{\pi_2})|_{\max}} \tag{20}$$

When $j \geq i > 2$:

$$
\begin{aligned}
&\delta(t)^{\text{sup\_min}}_{u\_s\_i-1\_i} \\
&= \frac{\epsilon_{\max}}{|c(p^c_{\pi_1} + \cdots + p^c_{\pi_{i-1}} - p^c_{\pi_j})\nabla_{\hat{\mathbf{x}}}(\mathbf{z}_{\pi_{i-1}} - \mathbf{z}_{\pi_i})|_{\max}} \\
&< \frac{\epsilon_{\max}}{|cp^c_{\pi_1} \nabla_{\hat{\mathbf{x}}}(\mathbf{z}_{\pi_{i-1}} - \mathbf{z}_{\pi_i})|_{\max}}
\end{aligned}
\tag{21}
$$

It is evident that $cp^c_{\pi_1}$ is a pivotal factor in controlling the upper bounds of all terms from $\delta(t)^{\text{sup\_min}}_{u\_s\_1\_2}$ to $\delta(t)^{\text{sup\_min}}_{u\_s\_j-1\_j}$. Increasing the value of $cp^c_{\pi_1}$ effectively reduces these upper bounds. We define $g(t)_{u\_s} = cp^c_{\pi_1}$, and its derivative is given by:

$$g'(t)_{u\_s} = \frac{p^c_{\pi_1}(B + c(\Delta_{\text{detach}}B - S))}{\Delta_{\text{detach}}B} > 0, \tag{22}$$

where, $\Delta_{\text{detach}} = \mathbf{z}_{\pi_1} - \mathbf{z}_{\pi_2}, B = \sum_{j=1}^{K} e^{c(\mathbf{z}_j - \mathbf{z}_{\pi_2})} = 1 + \sum_{j \neq \pi_2} e^{c(\mathbf{z}_j - \mathbf{z}_{\pi_2})}, S = \sum_{j=1}^{K}(\mathbf{z}_j - \mathbf{z}_{\pi_2})e^{c(\mathbf{z}_j - \mathbf{z}_{\pi_2})}, \Delta_{\text{detach}}B - S = \sum_{j=1}^{K}(\mathbf{z}_{\pi_1} - \mathbf{z}_{\pi_j})e^{c(\mathbf{z}_j - \mathbf{z}_{\pi_2})} > 0$.

This indicates that $g(t)_{u\_s}$ is monotonically increasing. As $t$ increases, $g(t)_{u\_s}$ grows accordingly; however, $t$ is constrained by floating-point underflow. We define $\lambda > 0$ as the threshold beyond which $e^{-\lambda} = 0$ due to underflow, with $\lambda$ taking values of 16.6355, 103.2789, and 744.4401 for 16-bit, 32-bit, and 64-bit floating-point representations, respectively. When $t(\mathbf{z}_{\pi_j} - \mathbf{z}_{\pi_1})/(\mathbf{z}_{\pi_1} - \mathbf{z}_{\pi_2}) < -\lambda$, $p^c_{\pi_j}$ becomes zero due to underflow. To ensure $p^c_{\pi_j}$ remains non-zero during the attack process, the maximum value of $t^*$ is bounded by $\frac{\lambda(\mathbf{z}_{\pi_1} - \mathbf{z}_{\pi_2})}{\mathbf{z}_{\pi_1} - \mathbf{z}_{\pi_j}}$. Thus, $t^*$ is defined as:

$$t^* = \frac{\lambda(\mathbf{z}_{\pi_1} - \mathbf{z}_{\pi_2})}{\mathbf{z}_{\pi_1} - \mathbf{z}_{\pi_j}} \tag{23}$$

## 6.2 Unsuccessful Attack Phase of Targeted Attacks

When $\mathbf{z}_{y_t} \neq \mathbf{z}_{\pi_1}$, Here, we assume $\mathbf{z}_{y_t} = \mathbf{z}_{\pi_j}$ for some $j \in \{2, \ldots, K\}$.

$$-CE(\mathbf{z}, y_t) = \log p_{y_t} = \log \frac{e^{\mathbf{z}_{y_t} - \mathbf{z}_{\pi_1}}}{\sum_{i=1}^{K} e^{\mathbf{z}_i - \mathbf{z}_{\pi_1}}} \tag{24}$$

$$-CE(c\mathbf{z}, y_t) = \log p^c_{y_t} = \log \frac{e^{c(\mathbf{z}_{y_t} - \mathbf{z}_{\pi_1})}}{\sum_{i=1}^{K} e^{c(\mathbf{z}_i - \mathbf{z}_{\pi_1})}} \tag{25}$$

$$-\nabla_{\hat{\mathbf{x}}} CE\left(c\mathbf{z}, y_t\right) = c\left(1 - p_{y_t}^c\right)\nabla_{\hat{\mathbf{x}}}\left(\mathbf{z}_{y_t} - \mathbf{z}_{\pi_1}\right) - \sum_{i \neq y_t} cp_i^c \nabla_{\hat{\mathbf{x}}}\left(\mathbf{z}_i - \mathbf{z}_{\pi_1}\right)$$

$$= c\left(1 - p_{\pi_j}^c\right)\nabla_{\hat{\mathbf{x}}}\left(\mathbf{z}_{\pi_j} - \mathbf{z}_{\pi_1}\right) - \sum_{i \neq j} cp_i^c \nabla_{\hat{\mathbf{x}}}\left(\mathbf{z}_{\pi_i} - \mathbf{z}_{\pi_1}\right)$$

$$= c\left(1 - p_{\pi_j}^c\right)\nabla_{\hat{\mathbf{x}}}\left(\mathbf{z}_{\pi_j} - \mathbf{z}_{\pi_{j-1}} + ... + \mathbf{z}_{\pi_2} - \mathbf{z}_{\pi_1}\right)$$

$$- cp_{\pi_2}^c \nabla_{\hat{\mathbf{x}}}\left(\mathbf{z}_{\pi_2} - \mathbf{z}_{\pi_1}\right)$$

$$- cp_{\pi_3}^c \nabla_{\hat{\mathbf{x}}}\left(\mathbf{z}_{\pi_3} - \mathbf{z}_{\pi_2} + \mathbf{z}_{\pi_2} - \mathbf{z}_{\pi_1}\right)$$

$$- ...$$

$$- cp_{\pi_K}^c \nabla_{\hat{\mathbf{x}}}\left(\mathbf{z}_{\pi_K} - \mathbf{z}_{\pi_{K-1}} + ... + \mathbf{z}_{\pi_2} - \mathbf{z}_{\pi_1}\right)$$

$$= c(1 - p_{\pi_j}^c - p_{\pi_2}^c - ... - p_{\pi_K}^c)\nabla_{\hat{\mathbf{x}}}\left(\mathbf{z}_{\pi_2} - \mathbf{z}_{\pi_1}\right)$$

$$+ c(1 - p_{\pi_j}^c - p_{\pi_3}^c - ... - p_{\pi_K}^c)\nabla_{\hat{\mathbf{x}}}\left(\mathbf{z}_{\pi_3} - \mathbf{z}_{\pi_2}\right)$$

$$+ ...$$

$$+ c(1 - p_{\pi_j}^c - p_{\pi_j}^c - ... - p_{\pi_K}^c)\nabla_{\hat{\mathbf{x}}}\left(\mathbf{z}_{\pi_j} - \mathbf{z}_{\pi_{j-1}}\right)$$

$$- c(p_{\pi_{j+1}}^c + ... + p_{\pi_K}^c)\nabla_{\hat{\mathbf{x}}}\left(\mathbf{z}_{\pi_{j+1}} - \mathbf{z}_{\pi_j}\right)$$

$$- ...$$

$$- cp_{\pi_K}^c \nabla_{\hat{\mathbf{x}}}\left(\mathbf{z}_{\pi_K} - \mathbf{z}_{\pi_{K-1}}\right)$$

$$= c(p_{\pi_1}^c - p_{\pi_j}^c)\nabla_{\hat{\mathbf{x}}}\left(\mathbf{z}_{\pi_2} - \mathbf{z}_{\pi_1}\right)$$

$$+ c(p_{\pi_1}^c + p_{\pi_2}^c - p_{\pi_j}^c)\nabla_{\hat{\mathbf{x}}}\left(\mathbf{z}_{\pi_3} - \mathbf{z}_{\pi_2}\right)$$

$$+ ...$$

$$+ c(p_{\pi_1}^c + ... + p_{\pi_{j-1}}^c - p_{\pi_j}^c)\nabla_{\hat{\mathbf{x}}}\left(\mathbf{z}_{\pi_j} - \mathbf{z}_{\pi_{j-1}}\right)$$

$$- c(p_{\pi_{j+1}}^c + ... + p_{\pi_K}^c)\nabla_{\hat{\mathbf{x}}}\left(\mathbf{z}_{\pi_{j+1}} - \mathbf{z}_{\pi_j}\right)$$

$$- ...$$

$$- cp_{\pi_K}^c \nabla_{\hat{\mathbf{x}}}\left(\mathbf{z}_{\pi_K} - \mathbf{z}_{\pi_{K-1}}\right) \tag{26}$$

where $c = \frac{t}{\Delta_{\text{detach}}}$ is a scale factor, $t > 0$, $\Delta_{\text{detach}} = \mathbf{z}_{\pi_1} - \mathbf{z}_{\pi_2} > 0$, and $p_{\pi_j}^c = e^{c\left(\mathbf{z}_{\pi_j} - \mathbf{z}_{\pi_1}\right)} / \sum_{j=1}^{K} e^{c\left(\mathbf{z}_{\pi_j} - \mathbf{z}_{\pi_1}\right)}$.

$$\mathbf{z}_{y_t} - \max_{i \neq y_t} \mathbf{z}_i = \mathbf{z}_{\pi_j} - \mathbf{z}_{\pi_1} = \left(\mathbf{z}_{\pi_j} - \mathbf{z}_{\pi_{j-1}}\right) + ... + \left(\mathbf{z}_{\pi_2} - \mathbf{z}_{\pi_1}\right) \tag{27}$$

Based on the primary objective of the targeted attack in the Unsuccessful Attack Phase, as specified in Equation equation 27, the gradients $\nabla_{\hat{\mathbf{x}}}(\mathbf{z}_{\pi_2} - \mathbf{z}_{\pi_1})$, $\nabla_{\hat{\mathbf{x}}}(\mathbf{z}_{\pi_3} - \mathbf{z}_{\pi_2})$, ..., $\nabla_{\hat{\mathbf{x}}}(\mathbf{z}_{\pi_j} - \mathbf{z}_{\pi_{j-1}})$ are critical components for gradient computation accuracy. To enhance the precision of these gradient computations, it is necessary to simultaneously minimize the upper bounds of the relative errors associated with the terms $|c(p_{\pi_1}^c - p_{\pi_j}^c)\nabla_{\hat{\mathbf{x}}}(\mathbf{z}_{\pi_2} - \mathbf{z}_{\pi_1})|$, $|c(p_{\pi_1}^c + p_{\pi_2}^c - p_{\pi_j}^c)\nabla_{\hat{\mathbf{x}}}(\mathbf{z}_{\pi_3} - \mathbf{z}_{\pi_2})|$, ..., $|c(p_{\pi_1}^c + p_{\pi_2}^c + \cdots + p_{\pi_{j-1}}^c - p_{\pi_j}^c)\nabla_{\hat{\mathbf{x}}}(\mathbf{z}_{\pi_j} - \mathbf{z}_{\pi_{j-1}})|$. We define $\delta(t)_{t\_u\_i\_i-1}^{\text{sup\_min}}$ as the upper bound of the relative error in $\nabla_{\hat{\mathbf{x}}}(\mathbf{z}_{\pi_i} - \mathbf{z}_{\pi_{i-1}})$ during the Unsuccessful Attack Phase of a targeted attack, where $i$ ranges from 2 to $j$:

When $i = 2$:

$$\delta(t)_{t\_u\_2\_1}^{\text{sup\_min}} = \frac{\epsilon_{\max}}{|c(p_{\pi_1}^c - p_{\pi_j}^c)\nabla_{\hat{\mathbf{x}}}(\mathbf{z}_{\pi_2} - \mathbf{z}_{\pi_1})|_{\max}} \tag{28}$$

When $j \geq i > 2$:

$$\delta(t)^{\text{sup\_min}}_{t\_u\_i\_i-1}$$

$$= \frac{\epsilon_{\max}}{|c(p^c_{\pi_1} + p^c_{\pi_2} + \cdots + p^c_{\pi_{i-1}} - p^c_{\pi_j})\nabla_{\hat{\mathbf{x}}}(\mathbf{z}_{\pi_i} - \mathbf{z}_{\pi_{i-1}})|_{\max}} \tag{29}$$

$$< \frac{\epsilon_{\max}}{|c(p^c_{\pi_1} - p^c_{\pi_j})\nabla_{\hat{\mathbf{x}}}(\mathbf{z}_{\pi_i} - \mathbf{z}_{\pi_{i-1}})|_{\max}}$$

It is evident that $c(p^c_{\pi_1} - p^c_{\pi_j})$ is a pivotal factor in controlling the upper bounds of all terms from $\delta(t)^{\text{sup\_min}}_{t\_u\_2\_1}$ to $\delta(t)^{\text{sup\_min}}_{t\_u\_j\_j-1}$. Increasing the value of $c(p^c_{\pi_1} - p^c_{\pi_j})$ effectively reduces these upper bounds. We define $g(t)_{t\_u} = c(p^c_{\pi_1} - p^c_{\pi_j})$, with its derivative given by:

$$g'(t)_{t\_u} = \frac{A(B + cD) + cS}{\Delta_{\text{detach}}A^2} > 0, \tag{30}$$

where $\Delta_{\text{detach}} = \mathbf{z}_{\pi_1} - \mathbf{z}_{\pi_2}$, $A = \sum_{i=1}^{K} e^{c(\mathbf{z}_{\pi_i} - \mathbf{z}_{\pi_1})}$, $B = 1 - e^{c(\mathbf{z}_{\pi_j} - \mathbf{z}_{\pi_1})} > 0$, $D = -ce^{c(\mathbf{z}_{\pi_j} - \mathbf{z}_{\pi_1})}(\mathbf{z}_{\pi_j} - \mathbf{z}_{\pi_1}) > 0$, and $S = -\sum_{i=1}^{K} e^{c(\mathbf{z}_{\pi_i} - \mathbf{z}_{\pi_1})}(\mathbf{z}_{\pi_i} - \mathbf{z}_{\pi_1}) > 0$. This indicates that $g(t)_{t\_u}$ is monotonically increasing. As $t$ increases, $g(t)_{t\_u}$ grows accordingly; Similar to the analysis of $t^*$ in Section 6.2, we define $t^*$ according to the following relation:

$$t^* = \frac{\lambda(\mathbf{z}_{\pi_1} - \mathbf{z}_{\pi_2})}{\mathbf{z}_{\pi_1} - \mathbf{z}_{\pi_j}} \tag{31}$$

### 6.3 Successful Attack Phase of Targeted Attacks

When $\mathbf{z}_{y_t} = \mathbf{z}_{\pi_1}$

$$-CE(\mathbf{z}, y_t) = \log p_{y_t} = \log \frac{e^{\mathbf{z}_{y_t} - \mathbf{z}_{\pi_2}}}{\sum_{i=1}^{K} e^{\mathbf{z}_i - \mathbf{z}_{\pi_2}}} \tag{32}$$

$$-CE(c\mathbf{z}, y_t) = \log p^c_{y_t} = \log \frac{e^{c(\mathbf{z}_{y_t} - \mathbf{z}_{\pi_2})}}{\sum_{i=1}^{K} e^{c(\mathbf{z}_i - \mathbf{z}_{\pi_2})}} \tag{33}$$

$$-\nabla_{\hat{\mathbf{x}}}CE(c\mathbf{z}, y_t) = c\left(1 - p^c_{\pi_1}\right)\nabla_{\hat{\mathbf{x}}}(\mathbf{z}_{\pi_1} - \mathbf{z}_{\pi_2}) - \sum_{i \neq 1} cp^c_{\pi_i}\nabla_{\hat{\mathbf{x}}}(\mathbf{z}_{\pi_i} - \mathbf{z}_{\pi_2})$$

$$= (1 - cp^c_{\pi_1})\nabla_{\hat{\mathbf{x}}}(\mathbf{z}_{\pi_1} - \mathbf{z}_{\pi_2})$$
$$- cp^c_{\pi_3}\nabla_{\hat{\mathbf{x}}}(\mathbf{z}_{\pi_3} - \mathbf{z}_{\pi_2})$$
$$- cp^c_{\pi_4}\nabla_{\hat{\mathbf{x}}}(\mathbf{z}_{\pi_4} - \mathbf{z}_{\pi_3} + \mathbf{z}_{\pi_3} - \mathbf{z}_{\pi_2})$$
$$- \ldots$$
$$- cp^c_{\pi_K}\nabla_{\hat{\mathbf{x}}}(\mathbf{z}_{\pi_K} - \mathbf{z}_{\pi_{K-1}} + \ldots + \mathbf{z}_{\pi_3} - \mathbf{z}_{\pi_2}) \tag{34}$$
$$= (1 - cp^c_{\pi_1})\nabla_{\hat{\mathbf{x}}}(\mathbf{z}_{\pi_1} - \mathbf{z}_{\pi_2})$$
$$- c(p^c_{\pi_3} + \ldots + p^c_{\pi_K})\nabla_{\hat{\mathbf{x}}}(\mathbf{z}_{\pi_3} - \mathbf{z}_{\pi_2})$$
$$- c(p^c_{\pi_4} + \ldots + p^c_{\pi_K})\nabla_{\hat{\mathbf{x}}}(\mathbf{z}_{\pi_4} - \mathbf{z}_{\pi_3})$$
$$- \ldots$$
$$- cp^c_{\pi_K}\nabla_{\hat{\mathbf{x}}}(\mathbf{z}_{\pi_K} - \mathbf{z}_{\pi_{K-1}})$$

where $y_t$ is a predefined target class, $y_t \in \{1, 2, \ldots, K\}$, and $y_t \neq y$, $c = \frac{t}{\Delta_{\text{detach}}}$ is a scale factor, $t > 0$, $\Delta_{\text{detach}} = |\mathbf{z}_{y_t} - \max_{i \neq y_t} \mathbf{z}_i|$, and $p^c_i = e^{c(\mathbf{z}_i - \mathbf{z}_{y_t})} / \sum_{j=1}^{K} e^{c(\mathbf{z}_j - \mathbf{z}_{y_t})}$.

$$\mathbf{z}_{y_t} - \max_{i \neq y_t} \mathbf{z}_i = \mathbf{z}_{\pi_1} - \mathbf{z}_{\pi_2} \tag{35}$$

Following the targeted attack objective defined in Equation equation 35, we establish that the gradient component $\nabla_{\hat{\mathbf{x}}}(\mathbf{z}_{\pi_1} - \mathbf{z}_{\pi_2})$ plays a pivotal role in unsuccessful attack scenarios. To optimize gradient computation accuracy, we specifically minimize the relative error $\delta(t)_{t\_s}$ in the gradient magnitude $|c(1 - p_{\pi_1}^c)\nabla_{\hat{\mathbf{x}}}(\mathbf{z}_{\pi_1} - \mathbf{z}_{\pi_2})|$.

$$\delta(t)_{t\_s}^{\text{sup\_min}} = \frac{\epsilon_{\max}}{|c(1 - p_{\pi_1}^c)\nabla_{\hat{\mathbf{x}}}(\mathbf{z}_{\pi_1} - \mathbf{z}_{\pi_2})|_{max}}, \tag{36}$$

The subsequent analysis follows the same methodology as section 3.1.1 this optimization is equivalent to finding the maximum values for the corresponding coefficients $c(1 - p_{\pi_1}^c)$,

$$g(t)_{t\_s} = c\left(1 - p_{\pi_1}^c\right) = c\left(1 - \frac{1}{B}\right) > 0 \tag{37}$$

Table 2: Comparing the proposed T-MIFPE loss ($\mathcal{L}^{\text{T-MIFPE}}$), against CE ($\mathcal{L}^{\text{ce}}$), and MIFPE($\mathcal{L}^{\text{MIFPE}}$) losses under targeted attacks (targeting the 9 closest incorrect classes). For each target, we use PGD-100 with the step-size schedule $\epsilon_i = \epsilon(1 + cos(\pi i/I))$, where $I = 100$ denotes the total iterations for each target and $i$ represents the current iteration. and momentum $\nu = 0.75$. Numbers in parentheses indicate the improvement w.r.t the CE baseline. The AutoAttack, calculated using an ensemble of attacks and a minimum of 4900 iterations.

| Defense method | Architecture | Clean | CE ($\mathcal{L}_{\text{target}}^{\text{ce}}$) 900 | MIFPE ($\mathcal{L}_{\text{target}}^{\text{MIFPE}}$) 900 | T-MIFPE ($\mathcal{L}_{\text{target}}^{\text{T-MIFPE}}$) 900 | AutoAttack 4900 | T-MIFPE-AutoAttack 4900 |
|---|---|---|---|---|---|---|---|
| **CIFAR-10, $\ell_\infty$, $\varepsilon = 8/255$** | | | | | | | |
| Uncovering limits Gowal et al. (2020) | WRN-70-16 | 91.10 | 66.46 | 65.87 (-0.59) | **65.85** (−0.61) | 65.87 | 65.82 |
| Fixing data augmentation Rebuffi et al. (2021) | WRN-106-16 | 88.50 | 65.56 | 64.66 (-0.90) | **64.62** (−0.94) | 64.58 | 64.58 |
| Fixing data augmentation Rebuffi et al. (2021) | WRN-70-16 | 88.54 | 65.17 | 64.26 (-0.91) | **64.24** (−0.93) | 64.20 | 64.20 |
| Uncovering limits Gowal et al. (2020) | WRN-28-10 | 89.48 | 64.00 | 62.81 (-1.19) | **62.80** (−1.20) | 62.76 | 62.76 |
| Adversarial weight perturbation Wu et al. (2021) | WRN-28-10 | 88.25 | 61.37 | 60.03 (-1.34) | **60.01** (−1.36) | 60.04 | 60.00 |
| Unlabeled data Carmon et al. (2019) | WRN-28-10 | 89.69 | 60.22 | 59.53 (-0.69) | **59.52** (−0.70) | 59.53 | 59.49 |
| HYDRA Sehwag et al. (2020) | WRN-28-10 | 88.98 | 57.88 | 57.14 (-0.74) | **57.17** (−0.77) | 57.14 | 57.14 |
| Overfitting Rice et al. (2020) | WRN-34-20 | 85.34 | 54.57 | 53.41 (-1.16) | **53.42** (−1.17) | 53.42 | 53.37 |
| Self-adaptive training Huang et al. (2020)‡ | WRN-34-10 | 83.48 | 54.43 | 53.31 (-1.12) | **53.30** (−1.13) | 53.34 | 53.22 |
| **CIFAR-100, $\ell_\infty$, $\epsilon = 8/255$** | | | | | | | |
| Adversarial weight perturbation Wu et al. (2020b) | WRN-34-10 | 60.38 | 30.15 | 28.83 (-1.32) | **28.82** (−1.34) | 28.86 | 28.81 |
| Pre-training Hendrycks et al. (2019) | WRN-28-10 | 59.23 | 29.87 | 28.50 (-1.37) | **28.41** (−1.46) | 28.42 | 28.39 |
| Progressive Hardening Sitawarin et al. (2020) | WRN-34-10 | 62.82 | 24.83 | 24.60 (-0.23) | **24.57** (−0.26) | 24.57 | 24.52 |
| Overfitting Rice et al. (2020) | RN-18 | 53.83 | 19.29 | 18.95 (-0.34) | **18.92** (−0.37) | 18.95 | 18.92 |
| **ImageNet, $\ell_\infty$, $\epsilon = 4/255$** | | | | | | | |
| Transfer Better Salman et al. (2020) | RN-50 | 64.02 | 35.84 | 34.65 (-1.19) | **34.64** (−1.20) | 34.96 | 34.62 |
| Robustness library Engstrom et al. (2019) | RN-50 | 62.56 | 30.08 | 29.36 (-0.72) | **29.22** (−0.86) | 29.22 | 29.16 |
| Fast adversarial training Wong et al. (2020) | RN-50 | 53.30 | 25.66 | 25.06 (-0.60) | **25.02** (−0.64) | 25.24 | 24.82 |
| Transfer Better Salman et al. (2020) | RN-18 | 52.92 | 26.58 | 25.24 (-1.34) | **25.22** (−1.36) | 25.26 | 25.16 |

## 7 LIMITATIONS

While our theoretical analysis comprehensively examines floating-point-induced relative errors in gradient-based attacks across untargeted and targeted scenarios (both unsuccessful and successful phases), the experimental improvements of T-MIFPE over MIFPE remain modest. This is primarily because our derived optimal scaling factor $t^*$ closely approximates 1 in the critical unsuccessful untargeted attack scenario, theoretically justifying MIFPE's empirical choice. Consequently, further gains from substituting $t^*$ for 1 are inherently limited, as substantial additional enhancements beyond this near-optimal baseline are unrealistic. Notably, these modest yet consistent improvements empirically corroborate the accuracy of our theoretical derivations, reinforcing their validity rather than diminishing the work's contributions.

## 8 COMPUTE RESOURCES

To ensure reproducibility of the experimental results presented in Section 4, we provide a detailed description of the compute resources used for all experiments. All experiments were conducted on a single NVIDIA GeForce RTX 2080 Ti GPU with 11 GB of GDDR6 memory. The system was equipped with 32 GB of RAM and 1 TB of SSD storage, running on a Linux-based operating system (Ubuntu 20.04). The software environment included Python 3.8, PyTorch 1.9, and standard libraries for implementing the PGD attack framework and loss functions (CE, MIFPE, and T-MIFPE).

Each experimental run, consisting of a PGD attack with 100 iterations on the MNIST, CIFAR-10, or CIFAR-100 datasets, was executed under $\ell_\infty$- or $\ell_2$-bounded threat models. The approximate execution time for a single run varied by dataset due to differences in image size and model complexity:

- **ImageNet**: Approximately 60–90 minutes per run for a single model and loss function combination.
- **CIFAR-10**: Approximately 15–60 minutes per run.
- **CIFAR-100**: Approximately 15–60 minutes per run.

These times account for the standardized configuration (100 iterations, momentum factor of 0.75, and a linearly decaying step-size schedule) and include data loading, model evaluation, and adversarial example generation. For each dataset and threat model, we conducted multiple runs to compare the five loss functions across different models, as detailed in Table 1. The total compute time for the experiments reported in the paper is estimated at approximately 150–200 hours of GPU time, depending on the specific configurations and models tested.

We used large language models to assist with polishing the manuscript, enhancing clarity and coherence.

