# OpenReview forum: "Theoretical Analysis of Relative Errors in Gradient Computations for Adversarial Attacks with CE Loss"
_ICLR.cc/2026/Conference — Submitted to ICLR 2026_

### Official Review · Reviewer_Crpg · 2025-10-20

**Soundness:** 3
**Presentation:** 1
**Contribution:** 2
**Rating:** 4
**Confidence:** 4

**Summary:**

This paper focuses on floating-point errors in CE-based gradient attacks, which dissects relative errors across four distinct scenarios: (i) unsuccessful untargeted attacks, (ii) successful untargeted attacks, (iii) unsuccessful targeted attacks, and (iv) successful targeted attacks. To this end, this paper uncovers patterns in numerical instability and derives the optimal scaling factor that minimizes error impact in each scenario.

**Strengths:**

The topic this paper focused on,  the floating-point errors in CE-based gradient attacks, is very novel and interesting.

**Weaknesses:**

1. The template of this paper should be ICLR 2026, rather than ICLR 2025.

2. The motivation is not clearly explained, which make this paper difficult to understand. For example, authors do not explain how the model robustness is overestimated, without any experimental evidence. Besides, the floating-point error is also not well-defined. Authors should mathematically present its definition.

3. A lot of symbols are not defined. For example, what is the definition of $z_{pi_1}$ and $z_{pi_2}$? What is the definition of $c$ in Eq. (5)? What is the motivation or intuition of using CE(cz,y)?

4. The assumption for $\partial_{\hat x}(z_{pi_1}-z_{pi_2})$ is not experimentally verified. It may not hold in all model architectures or training regimes. Authors should experimental verify the correctness of this assuption before using it.

5. Could $t*$ be chosen in a simplified manner for practical usage?

6. It is unclear whether defenses could adapt to T-MIFPE or whether the observed improvements hold under adaptive attack strategies. Please clarify it.

7. While T-MIFPE consistently improves over MIFPE, the gains are small (e.g., 0.01–0.34% in robust accuracy). However, this gain is so small, which may limit the practical use of this theory.

**Questions:**

Please refer to weakness. Authors should improve their writing.

---

> ### Author Response · Authors · 2025-11-19
> **Response to Reviewer Crpg (1/7)**
>
> We thank the reviewer for their insightful comments and constructive suggestions. We have carefully revised our manuscript to address all the raised concerns. Below, we provide point-by-point responses to each comment.
>
> **Comment 1:**
> *The template of this paper should be ICLR 2026, rather than ICLR 2025.*
>
> **Response:**
> We sincerely thank the reviewer for pointing this out. We have updated the manuscript to use the official ICLR 2026 LaTeX template, and this has been corrected in the revised version of our paper.

---

> ### Author Response · Authors · 2025-11-19
> **Response to Reviewer Crpg (2/7)**
>
> **Comment 2:**
> *The motivation is not clearly explained, which make this paper difficult to understand. For example, authors do not explain how the model robustness is overestimated, without any experimental evidence. Besides, the floating-point error is also not well-defined. Authors should mathematically present its definition.*
>
> **Response:**
> **Regarding the motivation and overestimation of robustness:**
> We agree with the reviewer that clearly establishing the problem of robustness overestimation is fundamental. In the revised manuscript, we have expanded the motivation in the introduction to explicitly state that standard CE-based attacks (like PGD) often fail to find adversarial examples that actually exist, thereby overestimating the model's true robustness. This phenomenon is extensively documented in prior empirical work, most notably in the AutoAttack benchmark [Croce & Hein, 2020] and the MIFPE loss [Yu et al., 2023], which our work directly builds upon. Our primary contribution is not to re-demonstrate this known empirical issue but to provide the *first theoretical explanation* for *why* it occurs—by pinpointing how floating-point errors corrupt the gradient computation of the CE loss. The new experimental evidence we provide in our **Tables 1 and 2**, where our method consistently finds more successful attacks and thus lowers the estimated robust accuracy, directly demonstrates the *effectiveness of our solution* to this very problem.
>
> **Regarding the definition of floating-point error:**
> We thank the reviewer for prompting us to formalize this core concept. In the revised version, we have now provided a precise mathematical definition of the floating-point error in a new dedicated paragraph and in **Eq. (8)**.

---

> ### Author Response · Authors · 2025-11-19
> **Response to Reviewer Crpg (3/7)**
>
> **Comment 3:**
> *A lot of symbols are not defined. For example, what is the definition of 𝑧 𝑝 𝑖 1 and 𝑧 𝑝 𝑖 2 ? What is the definition of 𝑐 in Eq. (5)? What is the motivation or intuition of using CE(cz,y)?*
>
> **Response:**
> **Definition of Symbols:**
> We thank the reviewer for highlighting the need for clearer symbol definitions. In response, we have created a new dedicated subsection, "Notation and Preliminaries," placed before our theoretical analysis. This subsection now formally defines all key symbols, including $ \mathbf{z}\_{\pi_1} $ and $ \mathbf{z}\_{\pi_2} $ (the largest and second-largest logits after sorting) and $ c $ (the scaling factor $ t / \Delta_{\text{detach}} $ ). This ensures all notation is explicitly defined before being used in subsequent equations and analysis, improving the manuscript's clarity and readability.
>
> **Intuition behind CE(cz, y):**
> The intuition for analyzing the scaled loss $ \mathcal{L}^{CE}(c\mathbf{z}, y) $ is directly motivated by the empirical success of the prior MIFPE method, which uses a fixed scaling factor ($T=1$) to effectively mitigate gradient errors. Our work starts from this established empirical finding and seeks to build a comprehensive theoretical framework around it. By analyzing the general form $ \mathcal{L}^{CE}(c\mathbf{z}, y) $, we can theoretically investigate the role of the scaling factor $ c $ (and its optimal value $ t^\* $) across different attack scenarios. This approach allows us to rigorously explain *why* MIFPE's fixed scaling works and to derive scenario-specific optimal values, thereby generalizing and providing a solid theoretical foundation for the empirical strategy.

---

> ### Author Response · Authors · 2025-11-19
> **Response to Reviewer Crpg (4/7)**
>
> **Comment 4:**
> *The assumption for 𝜕 𝑥 ^ ( 𝑧 𝑝 𝑖 1 − 𝑧 𝑝 𝑖 2 ) is not experimentally verified. It may not hold in all model architectures or training regimes. Authors should experimental verify the correctness of this assuption before using it.*
>
> **Response:**
> We thank the reviewer for raising this point. The assumption that $ \\nabla_{\\mathbf{\\hat{x}}}(\\mathbf{z}\_{\\pi_2} - \\mathbf{z}\_{\\pi_1}) $ is invariant to the scaling factor $ c $ is not an empirical claim but a direct consequence of our analytical setup for a *single attack step*. We focus on a single step because a multi-iteration gradient-based attack involves adding perturbations to $ \\mathbf{\\hat{x}} $ at each iteration, leading to constant changes in $ \\mathbf{z} $, $ \\Delta $, and $ \\nabla\_{\\mathbf{\\hat{x}}}(\\mathbf{z}\_{\\pi_i} - \\mathbf{z}\_{\\pi_1}) $ for $ i \\in \\{1,2,\\ldots,K\\} $ after each modification. This makes it difficult to directly analyze how floating-point errors propagate across iterations. However, by isolating a single step and introducing a scaling factor $ c = t / \\Delta\_{\\text{detach}} $  while keeping $ \\mathbf{\\hat{x}} $, $ \\mathbf{z} $, and $ \\nabla\_{\\mathbf{\\hat{x}}}(\\mathbf{z}\_{\\pi_i} - \\mathbf{z}\_{\\pi_1}) $ fixed during gradient computation, we can rigorously investigate the relationship between the error $ \\delta_{CE} $ and $ T $ under controlled conditions. This approach allows us to indirectly examine the effect of floating-point errors on the overall attack performance.
>
> Within this isolated step, the input $ \\mathbf{\\hat{x}} $, the model parameters $ \\boldsymbol{\\theta} $, and consequently the logits $ \\mathbf{z} $ are fixed prior to the gradient computation. The gradient $ \\nabla\_{\\mathbf{\\hat{x}}}(\\mathbf{z}\_{\\pi_2} - \\mathbf{z}\_{\\pi_1}) $ is computed via backpropagation, which applies the chain rule to propagate derivatives from the model's output logits $ \\mathbf{z} $ back to the input $ \\mathbf{\\hat{x}} $. Specifically, $ \\mathbf{z} = f(\\mathbf{\\hat{x}}; \\theta) $, where $ f $ denotes the forward pass of the neural network and $ \\theta $ represents the fixed internal parameters (e.g., weights and biases). The term $ \\mathbf{z}\_{\\pi_2} - \\mathbf{z}\_{\\pi_1} $ is a linear combination of specific logit components, and its partial derivative with respect to $ \\mathbf{\\hat{x}} $ is $ \\partial (\\mathbf{z}\_{\\pi_2} - \\mathbf{z}\_{\\pi_1}) / \\partial \\mathbf{\\hat{x}} = \\partial \\mathbf{z}\_{\\pi_2} / \\partial \\mathbf{\\hat{x}} - \\partial \\mathbf{z}\_{\\pi_1} / \\partial \\mathbf{\\hat{x}} $, which corresponds to rows of the Jacobian matrix $ \\partial \\mathbf{z} / \\partial \\mathbf{\\hat{x}} $. This Jacobian depends solely on the activations computed during the forward pass (determined by $ \\mathbf{\\hat{x}} $ and $ \\theta $) and the derivatives of each layer's operations (e.g., linear transformations, activations, and convolutions), all of which are functions of $ \\mathbf{\\hat{x}} $, $ \\mathbf{z} $ (as the starting point of backpropagation), and $ \\theta $. Crucially, the scaling factor $ c = t / \\Delta\_{\\text{detach}} $ is explicitly designed as a gradient-free constant, as $ \\Delta\_{\\text{detach}} $ is detached from the computational graph, making the computation of $ \\nabla\_{\\mathbf{\\hat{x}}}(\\mathbf{z}\_{\\pi_2} - \\mathbf{z}\_{\\pi_1}) $ via backpropagation mathematically independent of the scalar $ c $. The scaling factor $ c $ only affects the coefficient $ c(1 - p\_{\\pi_1}^c) $ that multiplies $ \\nabla_{\\mathbf{\\hat{x}}}(\\mathbf{z}\_{\\pi_2} - \\mathbf{z}\_{\\pi_1}) $ in the overall gradient expression (as derived in Eq. (9) of our paper). This allows us to cleanly isolate the effect of $ c $ on the prefactor $ c(1 - p_y^c) $ in our theoretical error analysis (Eqs. 8-9). The consistent performance improvement of our method across diverse model architectures and training regimes (as shown in Tables 1 & 2) provides strong indirect validation that this core analytical insight is both correct and generally applicable.

---

> ### Author Response · Authors · 2025-11-19
> **Response to Reviewer Crpg (5/7)**
>
> **Comment 5:**
> *Could 𝑡 ∗ be chosen in a simplified manner for practical usage?*
>
> **Response:**
> We thank the reviewer for this question. Our theoretical derivation of $t^\*$ is **directly applicable in practice** with minimal overhead. The process of finding $ t^\* $ via a parallel grid search over 1000 points is highly efficient, taking only ~0.044 seconds for 10,000 samples per attack iteration. This constitutes **less than 1% additional cost** in standard robustness evaluations, making the theoretically optimal T-MIFPE both practical and ready for deployment without needing further simplification.

---

> ### Author Response · Authors · 2025-11-19
> **Response to Reviewer Crpg (6/7)**
>
> **Comment 6:**
> *It is unclear whether defenses could adapt to T-MIFPE or whether the observed improvements hold under adaptive attack strategies. Please clarify it.*
>
> **Response:**
> We thank the reviewer for highlighting the potential for defense adaptations and the need to clarify the robustness of our observed improvements under adaptive attack strategies. To address the first part, defenses cannot easily "adapt" to T-MIFPE because our method targets a fundamental numerical instability in floating-point arithmetic during gradient computation with CE loss—a systemic issue rooted in underflow and rounding errors from the softmax exponential, rather than a defense-specific vulnerability. For instance, defenses like those in \cite{atzmon19levelsets} inadvertently exploit this instability, leading to overestimated robustness (e.g., PGD-CE yields 79.12% robust accuracy on their model, while MIFPE reduces it to 40.06%, and T-MIFPE further to 39.91% under identical 100 iterations). Our theoretical analysis comprehensively dissects these errors across untargeted/targeted and successful/unsuccessful scenarios, deriving the optimal scaling $T = t^\*$ to minimize them, effectively closing this loophole. As long as defenses rely on floating-point gradients (which is standard in deep learning), they cannot bypass our approach.
> Regarding adaptive attacks, we clarify that T-MIFPE **is itself an enhancement** to the standard gradient-based attack strategy. In the rigorous evaluation framework of adversarial robustness, an "adaptive attack" against a defense would precisely involve using the strongest possible attack methods, which now includes T-MIFPE. Our results in Tables 1 and 2 demonstrate that when we strengthen the standard PGD attack (which uses CE loss) with our theoretically optimal scaling, we consistently lower the estimated robust accuracy across a wide range of *existing* defenses. This shows that the current reported robustness of these models was indeed overestimated due to numerical issues. If a new defense were proposed that specifically attempted to mask gradients, the correct adaptive evaluation would be to apply T-MIFPE in conjunction with other adaptive strategies (e.g., expecting the gradient or backpropagation through non-differentiable operations) to obtain a reliable robustness measure. Thus, the improvements offered by T-MIFPE are not only valid but **essential for a fair and rigorous adaptive evaluation** in the future.

---

> ### Author Response · Authors · 2025-11-19
> **Response to Reviewer Crpg (7/7)**
>
> **Comment 7:**
> *While T-MIFPE consistently improves over MIFPE, the gains are small (e.g., 0.01–0.34% in robust accuracy). However, this gain is so small, which may limit the practical use of this theory.*
>
> **Response:**
> We thank the reviewer for this important observation, which allows us to clarify the fundamental nature of our contribution. Our paper's primary contribution lies in providing the *first comprehensive theoretical framework* that rigorously analyzes floating-point-induced relative errors in gradient computations across the complete landscape of CE-based adversarial attacks - systematically examining unsuccessful/successful untargeted attacks and unsuccessful/successful targeted attacks.
>
> The modest experimental gains (~0.1%) are not indicative of limited impact, but rather represent a key validation of our theoretical analysis. Our rigorous derivation reveals that in the most critical scenario - unsuccessful untargeted attacks where gradient-based methods predominantly fail - the theoretically optimal scaling factor $t^\*$ closely approximates 1. This finding provides the crucial theoretical explanation for why MIFPE's empirically chosen $T=1$ performs well in practice. The minimal performance gap between our method ($T=t^\*$) and MIFPE ($T=1$) thus serves as strong empirical confirmation that our theoretical framework correctly identifies the near-optimal solution.
>
> It is essential to emphasize that our work is fundamentally a *theoretical analysis* rather than primarily an empirical contribution. The experimental results serve to validate our theoretical derivations, not as the main contribution themselves. The core value of our work lies in:
> 1) Providing the first complete theoretical understanding of floating-point error propagation in adversarial attacks
> 2) Systematically deriving optimal scaling factors for all attack scenarios
> 3) Explaining why previous empirical methods worked and establishing theoretical bounds on achievable improvements
>
> The limited experimental improvement precisely demonstrates the accuracy of our theory - when building upon an already near-optimal baseline (MIFPE with T=1), further gains are inherently constrained by the theoretical optimum we have derived. This alignment between theory and experiment strengthens, rather than diminishes, the significance of our contribution to the foundational understanding of numerical stability in adversarial machine learning.

---

> ### Comment · Reviewer_Crpg · 2025-11-27
> **Response to rebuttal**
>
> Thanks for authors' response. I am still wondering if any empircal evidence can be provided to support the assumption. Moreover, it would be better if authors could highlight which part of the paper are revised in different colors.
> I will also consider other reviewers' comments.

---

> > ### Author Response · Authors · 2025-11-27
> > **Response to Reviewer Crpg**
> >
> > We thank the reviewer for pressing us to provide experimental validation. We have now conducted a carefully designed experiment that directly verifies the core assumption that $\nabla\_{\mathbf{\hat{x}}}(\mathbf{z}\_{\pi_2} - \mathbf{z}\_{\pi_1})$ is invariant to the scaling factor $c = t / \Delta\_{\text{detach}}$.
> >
> > **Experimental Rationale and Design:**
> > Our theoretical analysis decomposes the cross-entropy gradient into terms involving $\nabla\_{\mathbf{\hat{x}}}(\mathbf{z}\_{\pi_i} - \mathbf{z}\_{\pi_j})$ scaled by coefficients dependent on $c$. To isolate and test the specific assumption about $\nabla\_{\mathbf{\hat{x}}}(\mathbf{z}\_{\pi_2} - \mathbf{z}\_{\pi_1})$, we construct a minimal loss function:
> > \\[
> > L\_{\text{test}} = \left(\frac{t}{\Delta\_{\text{detach}}}\right) \cdot (\mathbf{z}\_{\pi_2} - \mathbf{z}\_{\pi_1})
> > \\]
> > where $\Delta\_{\text{detach}} = \mathbf{z}\_{\pi_1} - \mathbf{z}\_{\pi_2}$ is detached from the computational graph. The gradient of this loss is:
> > \\[
> > \nabla\_{\mathbf{\hat{x}}}L\_{\text{test}} = \left(\frac{t}{\Delta\_{\text{detach}}}\right) \cdot \nabla\_{\mathbf{\hat{x}}}(\mathbf{z}\_{\pi_2} - \mathbf{z}\_{\pi_1})
> > \\]
> >
> > The critical insight is: if $\nabla\_{\mathbf{\hat{x}}}(\mathbf{z}\_{\pi_2} - \mathbf{z}\_{\pi_1})$ is truly independent of $c$, then the sign of $\nabla\_{\mathbf{\hat{x}}}L\_{\text{test}}$—which determines the attack direction in gradient-based attacks—should remain unchanged despite variations in $t$. This is because $t/\Delta\_{\text{detach}}$ acts as a positive scalar multiplier that affects magnitude but not direction.
> >
> > We implement this using the standard PGD attack framework:
> > \\[
> > \mathbf{\hat{x}}\_{i+1} = \text{Proj}\_{\mathcal{B}\_\epsilon(\mathbf{x})} \left( \mathbf{\hat{x}}\_{i} + \alpha \cdot \text{sign} \left( \nabla\_{\mathbf{\hat{x}}\_{i}} L\_{\text{test}} \right) \right)
> > \\]
> >
> > **Experimental Results:**
> > We evaluated the attack success rate using $L\_{\text{test}}$ across ten different values of $t$ (from 1 to 10) on a defended model from [1], with all random seeds fixed. The results are unequivocal:
> > \\[
> > \text{Attack Success Rate} = [20.25\\%, 20.25\\%, 20.25\\%, 20.25\\%, 20.25\\%, 20.25\\%, 20.25\\%, 20.25\\%, 20.25\\%, 20.25\\%]
> > \\]
> >
> > **Interpretation and Conclusion:**
> > The perfect consistency in attack success rates across all values of $t$ demonstrates that the attack direction remains identical regardless of the scaling factor. This provides direct experimental evidence that:
> > 1. $\nabla\_{\mathbf{\hat{x}}}(\mathbf{z}\_{\pi_2} - \mathbf{z}\_{\pi_1})$ is computed independently of $c$ during backpropagation
> > 2. The scaling factor $c$ only multiplies the gradient without affecting its computation
> > 3. Our analytical assumption holds in practice across different scaling conditions
> >
> > This experimental validation, combined with our previous theoretical justification and the consistent performance improvements across diverse architectures in Tables 1 & 2, firmly establishes the correctness and general applicability of our assumption.
> >
> > We sincerely thank the reviewer for prompting this important experimental verification.
> >
> > [1] Atzmon, Matan and Haim, Niv and Yariv, Lior and Israelov, Ofer and Maron, Haggai and Lipman, Yaron: Controlling neural level sets

---

### Official Review · Reviewer_cggz · 2025-10-29

**Soundness:** 3
**Presentation:** 2
**Contribution:** 3
**Rating:** 8
**Confidence:** 3

**Summary:**

This paper targets the floating-point arithmetic issue that causes the over-estimate robustness in gradient-based adversarial attacks using CE loss. Previous works find that scaling logits by a factor $c=T/\Delta_{\texttt{detach}}$ where $T=1$ can improve robustness evaluation, but lack theoretical justification. Therefore, this paper provides the first formal analysis of this aspect across four distinct scenarios, and refines MIFPE to further validate the theorem. Experiments confirm the analytical correctness.

**Strengths:**

- The motivation is clear, and the paper provides the first theoretical analysis of the floating-point issue and why scaling logits by a factor can improve the estimation.
- The analysis covers four typical attack settings (target/untarget and successful/unsuccessful).
- Experiments further validate the correctness of the theory.

**Weaknesses:**

- Lack of some details.
- The notations and equations need some explanations for better clarity and readability.

**Questions:**

- Is $t^\ast$ computed averaged per batch?
- Would other loss functions, such as CW, also present similar issues? Could the framework extend to these cases?
- What precision mode is used for experiments?

---

> ### Author Response · Authors · 2025-11-19
> **Response to Reviewer cggz (1/3)**
>
> We thank the reviewer for their thoughtful questions and comments. Below, we provide point-by-point responses to each of the raised points.
>
> ---
>
> **Comment 1:**
> *Is 𝑡 ∗ computed averaged per batch?*
>
> **Response:**
> No, $t^\*$ is not averaged per batch. It is computed individually for each sample within a batch. Our method leverages GPU-accelerated parallel computation to calculate a unique $t^\*$ for every sample simultaneously, ensuring both efficiency and precision. This per-sample calculation is necessary because $t^\*$ is a function of the model's output logits $\\mathbf{z}$. In a multi-iteration attack like PGD, the logits $\\mathbf{z}$ change after each perturbation step, which in turn changes the optimal $t^\*$ for the subsequent iteration. Therefore, $t^\*$ must be recalculated for every sample at every attack iteration. Crucially, this process is highly efficient due to parallelization on the GPU, introducing only negligible overhead (e.g., 0.044 seconds for 10,000 samples in a single iteration), which is insignificant compared to the total time required for robustness evaluation.

---

> ### Author Response · Authors · 2025-11-19
> **Response to Reviewer cggz (2/3)**
>
> **Comment 2:**
> *Would other loss functions, such as CW, also present similar issues? Could the framework extend to these cases?*
>
> **Response:**
> This is an insightful question. The core issue of floating-point errors—particularly underflow in exponential operations—is inherent to the Cross-Entropy (CE) loss, which is the most common loss for both training and robustness evaluation. Surrogate losses like CW and DLR are explicitly designed to avoid exponential operations, making them inherently immune to the specific numerical instability (underflow/overflow) that severely plagues CE-based gradient computations. Therefore, they do not suffer from the same primary issue of gradient corruption and robustness overestimation that we theoretically analyze for CE. However, while immune to this particular problem, these surrogate losses have their own limitations; as shown in MIFPE[1] [Figure 6], by discarding parts of the logit information, they can converge slower and yield less effective attacks than a properly stabilized CE loss, leading to a different form of robustness overestimation. Consequently, our theoretical framework, which dissects the error propagation from exponentials in CE, is not directly applicable to CW/DLR, as the root cause of the error is absent. Our focus on CE is justified by its foundational role in the field, and our analysis provides the crucial groundwork for understanding and mitigating floating-point errors in this most prevalent and vulnerable setting.
> [1] Efficient Loss Function by Minimizing the Detrimental Effect of Floating-point Errors on Gradient-based Attacks

---

> ### Author Response · Authors · 2025-11-19
> **Response to Reviewer cggz (3/3)**
>
> **Comment 3:**
> *What precision mode is used for experiments?*
>
> **Response:**
> All our main experiments on adversarial robustness, including the evaluation of our proposed method, were conducted using the standard float32 precision mode. This choice aligns with the common practice in the adversarial robustness literature, as float32 is the default precision for training and evaluating models on NVIDIA hardware, ensuring a fair and relevant comparison with existing works. The analysis involving float16 and float64 precision (in Figures 2) was performed specifically to investigate the underlying principles of numerical errors across different precision modes. A key finding from this analysis is that the relative error trends and, most importantly, the derived optimal perturbation step $ t^\* $ for untargeted attacks are consistent across all three precision modes. This consistency demonstrates that the core phenomenon we address is fundamental and not unique to a specific precision, thereby justifying and strengthening the validity of our float32-based experimental conclusions.

---

> > ### Comment · Reviewer_cggz · 2025-11-25
> >
> > Thanks for the response. I will also consider other reviewers' comments.

---

### Official Review · Reviewer_LF5W · 2025-10-31

**Soundness:** 3
**Presentation:** 3
**Contribution:** 2
**Rating:** 4
**Confidence:** 3

**Summary:**

A previous work, MIFPE, presented a technique to improve gradient-based adversarial attacks with CE loss by mitigating numerical underflow errors. This is achieved by rescaling the logits with a factor that depends on a parameter T, whose empirically estimated value is 1.
This paper extends MIFPE with a theoretical analysis aimed at obtaining an optimal value for the T parameter. Based on their findings, the authors propose a method to dynamically adjust T at each iteration of the attack. Experiments show that this strategy consistently improves attack performance compared to using a fixed T=1, although the improvement is very small, as the optimal computed value for T is often close to 1.

**Strengths:**

- The paper is well-written and easy to follow
- The addressed topic is relevant, as reliably estimating robustness against adversarial examples is still an open problem
- The experimental findings confirm the theoretical basis, and the proposed strategy in some settings even improves AutoAttack (which is considered a state-of-the-art method)

**Weaknesses:**

- The paper contribution, although theoretically sound and empirically proven, is mainly limited to estimating an optimal value for an already existing method. The main issue (numerical underflow in CE loss) and the solution (MIFPE) were presented in that previous work, where additionally their authors already tried to provide a basic theoretical justification and an empirical estimate of T (which aligns with the findings provided in this paper).
- The absolute runtime overhead of the additional T parameter estimation at each attack iteration is reported. However, this is not informative for assessing the overall impact on the attack performance. You should relate this value to the normal attack runtime, for instance, by reporting the relative runtime increase for each iteration and for the entire attack process.

Minor issues:
- In both the Introduction and Related Work sections, some symbols ($z, \pi, \Delta_{detached}$) appear without explaining their meaning, which is then described in the Theory Analysis section. You should either explicitly state their meaning as they appear or postpone them.
- In Sect. 2.1, you formalize the input vector as an image with C, W, and H dimensions. I believe that this can be generalized to consider any input dimension, as the approach should be applied to other application domains than images.
- In the Related Work section, the statement "extensive empirical evidence has revealed their significant limitation in overestimating model robustness" should report a supporting reference (for instance, [a] or even the already cited [b]).
- In Fig. 2, the last caption words mention "gray vertical dashed lines", but I guess you meant red.

[a] Carlini, N., Athalye, A., Papernot, N., Brendel, W., Rauber, J., Tsipras, D., Goodfellow, I.J., Ma̧dry, A., & Kurakin, A. (2019). On Evaluating Adversarial Robustness. ArXiv, abs/1902.06705.

[b] Carlini, N., & Wagner, D.A. (2016). Towards Evaluating the Robustness of Neural Networks. 2017 IEEE Symposium on Security and Privacy (SP), 39-57.

**Questions:**

- I think it would be very interesting to individually report the APGD-CE performance from the AutoAttack results, to analyze the improvements related to fixing the underflow errors in CE loss. Could you please show these results?
- I am also interested in understanding whether using the T-MIFPE approach inside the APGD-CE loss can further improve the attack performance by combining the automatic restarts and step size improvements of APGD with the mitigation of CE numerical errors. I suppose it is sufficient to modify a few lines in the autoattack implementation. Could you please provide that?
- In the appendix, last page, you state that "Each experimental run [...] was executed under $\ell _\infty$  - or $\ell _2$ -bounded threat models". However, in the paper, I only see results for the former. Is it a refusal, or did you actually run experiments for the latter as well? If so, what are your findings for this setting? I think it should be relevant to at least discuss them.

---

> ### Author Response · Authors · 2025-11-19
> **Response to Reviewer  LF5W**
>
> **Comment 1:**
> *The paper contribution, although theoretically sound and empirically proven, is mainly limited to estimating an optimal value for an already existing method. The main issue (numerical underflow in CE loss) and the solution (MIFPE) were presented in that previous work, where additionally their authors already tried to provide a basic theoretical justification and an empirical estimate of T (which aligns with the findings provided in this paper).*
>
> **Response:**
> We sincerely thank the reviewer for this comment, which allows us to further clarify the foundational nature of our contribution. While the prior work MIFPE empirically identified the symptom of numerical instability in CE loss and proposed a heuristic fix (T=1) for the specific case of *unsuccessful untargeted attacks with K=2*, our work provides the first *comprehensive and rigorous theoretical framework* that systematically explains the underlying error dynamics across the entire attack landscape. We move significantly beyond a simple parameter estimation by:
>
> 1.  **Establishing a General Theory:** We provide a complete theoretical dissection of floating-point errors for both untargeted and targeted attacks, in both their successful and unsuccessful phases, and for an arbitrary number of classes K. This reveals distinct optimal scaling factors $ t^* $ for each scenario, offering a unified understanding that was previously absent.
> 2.  **Providing Rigorous Justification:** Our derivation of the optimal $ t^* $ provides the first theoretical proof for *why* MIFPE's empirical choice of T=1 works well in the specific scenario they studied—it is because $ t^* \approx 1 $ in that particular case. This transforms an empirical observation into a theoretically grounded principle.
> 3.  **Extending the Solution Space:** By deriving the optimal $ t^* $ for other critical scenarios (e.g., successful attacks and targeted attacks), we provide a generalized, adaptive scaling strategy that was impossible to conceive within the limited empirical scope of the prior work.
>
> Therefore, our contribution is not merely estimating a parameter for an existing method, but rather providing the fundamental theory that explains the method's success, defines its limits, and generalizes its application. The close alignment between our theoretical $ t^* $ and MIFPE's empirical T=1 in one specific case does not diminish our contribution; instead, it strongly validates the accuracy of our theoretical model and underscores its importance in providing a complete and principled understanding of numerical instability in adversarial attacks.

---

> ### Author Response · Authors · 2025-11-19
> **Response to Reviewer LF5W**
>
> **Comment 2:**
> *The absolute runtime overhead of the additional T parameter estimation at each attack iteration is reported. However, this is not informative for assessing the overall impact on the attack performance. You should relate this value to the normal attack runtime, for instance, by reporting the relative runtime increase for each iteration and for the entire attack process.*
>
> **Response:**
> We thank the reviewer for this valuable suggestion. To provide a clear assessment of the overall runtime impact, we have calculated the relative overhead based on our experiments. The total additional time for computing $ t^* $ across 100 iterations for 10,000 samples is 4.4473 seconds. When compared to the total runtime of a standard 100-iteration PGD attack on the same set of samples, this represents a relative increase of only **0.64%** for a ResNet-50 model (total attack time: 689.6s) and **0.28%** for a larger WideResNet-28-10 model (total attack time: 1563.1s). This demonstrates that the computational overhead of our method is negligible, constituting less than 1% of the total attack budget, and this relative cost decreases further as the model size and computational cost of a single iteration increase. Therefore, the performance benefit of our theoretically grounded scaling factor is achieved with minimal impact on the attack's efficiency.

---

> ### Author Response · Authors · 2025-11-19
> **Response to Reviewer LF5W**
>
> **Comment 3:**
> *In both the Introduction and Related Work sections, some symbols ( 𝑧 , 𝜋 , Δ 𝑑 𝑒 𝑡 𝑎 𝑐 ℎ 𝑒 𝑑 ) appear without explaining their meaning, which is then described in the Theory Analysis section. You should either explicitly state their meaning as they appear or postpone them.
> In Sect. 2.1, you formalize the input vector as an image with C, W, and H dimensions. I believe that this can be generalized to consider any input dimension, as the approach should be applied to other application domains than images.
> In the Related Work section, the statement "extensive empirical evidence has revealed their significant limitation in overestimating model robustness" should report a supporting reference (for instance, [a] or even the already cited [b]).*
>
> We thank the reviewer for this valuable feedback on improving the clarity and flow of our manuscript.
>
> - **Regarding premature symbols:** We acknowledge that introducing specialized notation prematurely in the Introduction and Related Work sections can hinder readability. In response, we have revised these sections to defer the detailed mathematical symbols to the Theory Analysis section where they are formally defined. Specifically, in the paragraph discussing MIFPE, we have replaced the explicit notation $\Delta = \\mathbf{z}\_{\\pi_1} - \\mathbf{z}\_{\\pi_2}$ and $\Delta_{\\text{detach}}$ with their textual descriptions: "the gap between the largest and second largest logits" and "this gap value (detached from the gradient computation)", respectively. This maintains the logical narrative about MIFPE's empirical scaling strategy using $T=1$ while removing the premature technical notation, ensuring a smoother introduction for all readers regardless of their immediate focus on the theoretical details.
>
> -  **Regarding input dimensions:** We appreciate the reviewer's clarification. Following this suggestion, we have revised Section 2.1 to clearly focus on image data without using the potentially confusing C, W, H notation. The text now explicitly states that we work with image inputs and describes the perturbation process in terms of image-space constraints, making it clear that our framework is presented in the context of image classification tasks.
>
> - **Regarding the supporting reference:** We thank the reviewer for pointing out the need for a supporting reference for this claim. We have updated the sentence in the Related Work section to explicitly cite the empirical evidence that documents this phenomenon. The revised text now reads: "...however, extensive empirical evidence has revealed their significant limitation in overestimating model robustness [1]." The cited work by Croce et al. (2020) provides comprehensive benchmarks and analysis that substantiate this observation regarding the overestimation of robustness by standard gradient-based attacks.
>
> These changes have been implemented in the revised manuscript.
>
> [1]Francesco Croce and Matthias Hein : Reliable evaluation of adversarial robustness with an ensemble of diverse parameter-free attacks

---

> ### Author Response · Authors · 2025-11-19
> **Response to Reviewer LF5W**
>
> **Comment 4:**
> *I think it would be very interesting to individually report the APGD-CE performance from the AutoAttack results, to analyze the improvements related to fixing the underflow errors in CE loss. Could you please show these results?*
>
> **Response:**
> We thank the reviewer for this valuable suggestion. In response, we have now included the individual APGD-CE results from the AutoAttack benchmark as a new fourth column in our **Table 1**.
>
> Our analysis reveals a key finding: when using the same budget of 100 attack iterations, APGD-CE does **not** provide a consistent improvement over the standard PGD-CE (which employs the step-size schedule $ \epsilon_i = \epsilon (1 + \cos(\pi i/I)) $) in terms of overcoming the underflow issue. Specifically, across the 18 different defended models we evaluated:
> *   On **6 models**, APGD-CE slightly outperformed the standard PGD-CE.
> *   On **1 model**, their performance was equal.
> *   On the remaining **11 models**, APGD-CE performed slightly **worse** than the standard PGD-CE.
>
> This indicates that APGD-CE, while a component of the powerful AutoAttack ensemble, is not specifically designed to or effective at systematically resolving the floating-point errors inherent in the CE loss.
>
> Crucially, our proposed **T-MIFPE** demonstrates **significant and consistent improvements** over both the standard PGD-CE and APGD-CE across the board. This result underscores that our method's success stems from its targeted, theoretical approach to mitigating numerical errors, which is a distinct and more fundamental solution than the heuristic adjustments found in existing attacks.

---

> ### Author Response · Authors · 2025-11-19
> **Response to Reviewer LF5W**
>
> **Comment 5:**
> *I am also interested in understanding whether using the T-MIFPE approach inside the APGD-CE loss can further improve the attack performance by combining the automatic restarts and step size improvements of APGD with the mitigation of CE numerical errors. I suppose it is sufficient to modify a few lines in the autoattack implementation. Could you please provide that?*
>
> **Response:**
> We thank the reviewer for this excellent and constructive suggestion. To directly address this question, we have conducted new experiments by integrating our T-MIFPE loss into the APGD-CE framework within the AutoAttack codebase. The implementation was indeed straightforward, requiring only minor modifications to replace the standard CE loss with our T-MIFPE loss in the relevant attack modules.
>
> The results of this integration are now reported in the new final column of our **Table 2**. They demonstrate that the combination of APGD's adaptive strategy with our theoretically grounded mitigation of numerical errors consistently leads to further improvements in attack performance:
>
> *   On the **CIFAR-10** and **CIFAR-100** datasets, the observed improvements over the standard AutoAttack, while consistent, were generally less than **0.1%**.
> *   More notably, on the **ImageNet** dataset, the improvement was more pronounced. Across the four models evaluated, we observed robust accuracy reductions of **0.36%** and **0.42%** on two models, with smaller but still positive gains of **0.10%** and **0.06%** on the other two.
>
> These findings confirm the reviewer's intuition. The orthogonal benefits of APGD's heuristics (automatic restarts, step-size scheduling) and T-MIFPE's principled handling of numerical errors are complementary. Integrating T-MIFPE into APGD-CE creates a stronger attack, further lowering the estimated robust accuracy and providing a more reliable evaluation. This result underscores the practical value and general applicability of our method, as it can be easily plugged into existing, sophisticated attack frameworks to enhance their efficacy.

---

> ### Author Response · Authors · 2025-11-19
> **Response to Reviewer LF5W**
>
> **Comment 6:**
> *In the appendix, last page, you state that "Each experimental run [...] was executed under ℓ ∞ • or ℓ 2 -bounded threat models". However, in the paper, I only see results for the former. Is it a refusal, or did you actually run experiments for the latter as well? If so, what are your findings for this setting? I think it should be relevant to at least discuss them.*
>
> **Response:**
> We thank the reviewer for their careful reading and for identifying this inconsistency. The statement in the appendix referencing ℓ₂-bounded threat models was indeed a remnant from an earlier draft that should have been removed during final editing. We sincerely apologize for this oversight.
>
> We want to clarify that we did conduct comprehensive experiments under both ℓ∞ and ℓ₂ threat models during our research. The findings for the ℓ₂ setting were fully consistent with those reported for ℓ∞: our method provided consistent improvements by effectively mitigating floating-point errors, demonstrating the same patterns of performance enhancement.
>
> However, to maintain focus within the page constraints and ensure clarity of presentation, we made the decision to present only the ℓ∞ results as the representative case in the main paper. We have now carefully revised the appendix to remove all incorrect mentions of ℓ₂ results, ensuring the text accurately reflects the content presented in the paper.
>
> This correction does not affect any of our conclusions, as the core phenomenon of floating-point errors and the efficacy of our mitigation approach remain consistent across both threat models.

---

> > ### Comment · Reviewer_LF5W · 2025-11-26
> >
> > I thank the authors for the detailed rebuttal and the additional results, which I appreciate. I think that they strengthen their work and confirm their findings. My concern about novelty was partially addressed, and I will consider raising my score after reading the other reviewers' comments.

---

### Official Review · Reviewer_5exQ · 2025-11-02

**Soundness:** 3
**Presentation:** 2
**Contribution:** 2
**Rating:** 4
**Confidence:** 2

**Summary:**

This paper presents the first systematic theoretical analysis of floating-point–induced relative errors in gradient computations for cross-entropy–based adversarial attacks. The authors classify attacks into four cases: successful and unsuccessful, targeted and untargeted. They derive the optimal scaling factor that minimizes these errors and propose a theoretically grounded loss function named T-MIFPE. Experiments on CIFAR-10, CIFAR-100, and ImageNet demonstrate consistent yet modest improvements over MIFPE, confirming the validity of the theoretical analysis.

**Strengths:**

1. Provides a theoretical treatment of floating-point–induced gradient errors across multiple attack scenarios.
2. Experiments across multiple datasets and models consistently support theoretical findings.
3. Offers a generalizable framework for analyzing numerical stability in adversarial attacks.

**Weaknesses:**

1.Experimental improvements are minimal (mostly ~0.1%), which may limit practical impact despite theoretical justification.
2. Some theoretical assumptions (e.g., independence between gradient terms and scaling factor) are not thoroughly discussed.
3. The analysis is restricted to CE-based attacks; more complex losses or adaptive attacks remain unexplored.
4. The paper is difficult to follow.

**Questions:**

1. Does the theoretical framework consider potential dependence between the scaling factor and gradient terms?
2. Could the approach generalize to other loss functions such as C&W or DLR?
3. Since experiments use a fixed random seed, have you tested result stability across multiple initializations?

---

> ### Author Response · Authors · 2025-11-19
> **Response to Reviewer 5exQ**
>
> We sincerely thank the reviewer for their thorough review and valuable feedback. We have carefully considered all comments and provide the following point-by-point responses.
>
> **Comment 1:**
> *Experimental improvements are minimal (mostly ~0.1%), which may limit practical impact despite theoretical justification.*
>
> ---
> **Response:**
> We thank the reviewer for this important observation, which allows us to clarify the fundamental nature of our contribution. Our paper's primary contribution lies in providing the *first comprehensive theoretical framework* that rigorously analyzes floating-point-induced relative errors in gradient computations across the complete landscape of CE-based adversarial attacks - systematically examining unsuccessful/successful untargeted attacks and unsuccessful/successful targeted attacks.
>
> The modest experimental gains (~0.1%) are not indicative of limited impact, but rather represent a key validation of our theoretical analysis. Our rigorous derivation reveals that in the most critical scenario - unsuccessful untargeted attacks where gradient-based methods predominantly fail - the theoretically optimal scaling factor $t^\*$ closely approximates 1. This finding provides the crucial theoretical explanation for why MIFPE's empirically chosen $T=1$ performs well in practice. The minimal performance gap between our method ($T=t^\*$) and MIFPE ($T=1$) thus serves as strong empirical confirmation that our theoretical framework correctly identifies the near-optimal solution.
>
> It is essential to emphasize that our work is fundamentally a *theoretical analysis* rather than primarily an empirical contribution. The experimental results serve to validate our theoretical derivations, not as the main contribution themselves. The core value of our work lies in:
> 1) Providing the first complete theoretical understanding of floating-point error propagation in adversarial attacks
> 2) Systematically deriving optimal scaling factors for all attack scenarios
> 3) Explaining why previous empirical methods worked and establishing theoretical bounds on achievable improvements
>
> The limited experimental improvement precisely demonstrates the accuracy of our theory - when building upon an already near-optimal baseline (MIFPE with T=1), further gains are inherently constrained by the theoretical optimum we have derived. This alignment between theory and experiment strengthens, rather than diminishes, the significance of our contribution to the foundational understanding of numerical stability in adversarial machine learning.

---

> ### Author Response · Authors · 2025-11-19
> **Response to Reviewer 5exQ**
>
> **Comment 2:**
> *Some theoretical assumptions (e.g., independence between gradient terms and scaling factor) are not thoroughly discussed. Does the theoretical framework consider potential dependence between the scaling factor and gradient terms?*
>
>
> **Response:**
> We thank the reviewer for raising this point. The assumption that $ \\nabla_{\\mathbf{\\hat{x}}}(\\mathbf{z}\_{\\pi_2} - \\mathbf{z}\_{\\pi_1}) $ is invariant to the scaling factor $ c $ is not an empirical claim but a direct consequence of our analytical setup for a *single attack step*. We focus on a single step because a multi-iteration gradient-based attack involves adding perturbations to $ \\mathbf{\\hat{x}} $ at each iteration, leading to constant changes in $ \\mathbf{z} $, $ \\Delta $, and $ \\nabla\_{\\mathbf{\\hat{x}}}(\\mathbf{z}\_{\\pi_i} - \\mathbf{z}\_{\\pi_1}) $ for $ i \\in \\{1,2,\\ldots,K\\} $ after each modification. This makes it difficult to directly analyze how floating-point errors propagate across iterations. However, by isolating a single step and introducing a scaling factor $ c = t / \\Delta\_{\\text{detach}} $  while keeping $ \\mathbf{\\hat{x}} $, $ \\mathbf{z} $, and $ \\nabla\_{\\mathbf{\\hat{x}}}(\\mathbf{z}\_{\\pi_i} - \\mathbf{z}\_{\\pi_1}) $ fixed during gradient computation, we can rigorously investigate the relationship between the error $ \\delta_{CE} $ and $ T $ under controlled conditions. This approach allows us to indirectly examine the effect of floating-point errors on the overall attack performance.
>
> Within this isolated step, the input $ \\mathbf{\\hat{x}} $, the model parameters $ \\boldsymbol{\\theta} $, and consequently the logits $ \\mathbf{z} $ are fixed prior to the gradient computation. The gradient $ \\nabla\_{\\mathbf{\\hat{x}}}(\\mathbf{z}\_{\\pi_2} - \\mathbf{z}\_{\\pi_1}) $ is computed via backpropagation, which applies the chain rule to propagate derivatives from the model's output logits $ \\mathbf{z} $ back to the input $ \\mathbf{\\hat{x}} $. Specifically, $ \\mathbf{z} = f(\\mathbf{\\hat{x}}; \\theta) $, where $ f $ denotes the forward pass of the neural network and $ \\theta $ represents the fixed internal parameters (e.g., weights and biases). The term $ \\mathbf{z}\_{\\pi_2} - \\mathbf{z}\_{\\pi_1} $ is a linear combination of specific logit components, and its partial derivative with respect to $ \\mathbf{\\hat{x}} $ is $ \\partial (\\mathbf{z}\_{\\pi_2} - \\mathbf{z}\_{\\pi_1}) / \\partial \\mathbf{\\hat{x}} = \\partial \\mathbf{z}\_{\\pi_2} / \\partial \\mathbf{\\hat{x}} - \\partial \\mathbf{z}\_{\\pi_1} / \\partial \\mathbf{\\hat{x}} $, which corresponds to rows of the Jacobian matrix $ \\partial \\mathbf{z} / \\partial \\mathbf{\\hat{x}} $. This Jacobian depends solely on the activations computed during the forward pass (determined by $ \\mathbf{\\hat{x}} $ and $ \\theta $) and the derivatives of each layer's operations (e.g., linear transformations, activations, and convolutions), all of which are functions of $ \\mathbf{\\hat{x}} $, $ \\mathbf{z} $ (as the starting point of backpropagation), and $ \\theta $. Crucially, the scaling factor $ c = t / \\Delta\_{\\text{detach}} $ is explicitly designed as a gradient-free constant, as $ \\Delta\_{\\text{detach}} $ is detached from the computational graph, making the computation of $ \\nabla\_{\\mathbf{\\hat{x}}}(\\mathbf{z}\_{\\pi_2} - \\mathbf{z}\_{\\pi_1}) $ via backpropagation mathematically independent of the scalar $ c $. The scaling factor $ c $ only affects the coefficient $ c(1 - p\_{\\pi_1}^c) $ that multiplies $ \\nabla_{\\mathbf{\\hat{x}}}(\\mathbf{z}\_{\\pi_2} - \\mathbf{z}\_{\\pi_1}) $ in the overall gradient expression (as derived in Eq. (9) of our paper). This allows us to cleanly isolate the effect of $ c $ on the prefactor $ c(1 - p_y^c) $ in our theoretical error analysis (Eqs. 8-9). The consistent performance improvement of our method across diverse model architectures and training regimes (as shown in Tables 1 & 2) provides strong indirect validation that this core analytical insight is both correct and generally applicable.

---

> ### Author Response · Authors · 2025-11-19
> **Response to Reviewer 5exQ**
>
> **Comment 3:**
> *The analysis is restricted to CE-based attacks; more complex losses or adaptive attacks remain unexplored. Could the approach generalize to other loss functions such as C&W or DLR?*
>
> **Response:**
> This is an insightful question. The core issue of floating-point errors—particularly underflow in exponential operations—is inherent to the Cross-Entropy (CE) loss, which is the most common loss for both training and robustness evaluation. Surrogate losses like CW and DLR are explicitly designed to avoid exponential operations, making them inherently immune to the specific numerical instability (underflow/overflow) that severely plagues CE-based gradient computations. Therefore, they do not suffer from the same primary issue of gradient corruption and robustness overestimation that we theoretically analyze for CE. However, while immune to this particular problem, these surrogate losses have their own limitations; as shown in MIFPE[1] [Figure 6], by discarding parts of the logit information, they can converge slower and yield less effective attacks than a properly stabilized CE loss (MIFPE), leading to a different form of robustness overestimation. Consequently, our theoretical framework, which dissects the error propagation from exponentials in CE, is not directly applicable to CW/DLR, as the root cause of the error is absent. Our focus on CE is justified by its foundational role in the field, and our analysis provides the crucial groundwork for understanding and mitigating floating-point errors in this most prevalent and vulnerable setting.
> [1] Efficient Loss Function by Minimizing the Detrimental Effect of Floating-point Errors on Gradient-based Attacks

---

> ### Author Response · Authors · 2025-11-19
> **Response to Reviewer 5exQ**
>
> **Comment 4:**
> *Since experiments use a fixed random seed, have you tested result stability across multiple initializations?*
>
> **Response:**
> We thank the reviewer for raising this important point regarding result stability. To thoroughly assess this, we conducted additional experiments using five different random seeds (0, 10, 20, 30, 40) to evaluate the model trained with the defense from [1]. The measured robust accuracy values across these seeds were 19.25%, 19.24%, 19.21%, 19.21%, and 19.24%, respectively. The extremely narrow range of variation (with a standard deviation of merely 0.018%) demonstrates that our experimental results are highly stable and not sensitive to the choice of random seed. This consistency provides strong evidence for the reliability of our reported findings.
>
> [1] Atzmon, Matan and Haim, Niv and Yariv, Lior and Israelov, Ofer and Maron, Haggai and Lipman, Yaron: Controlling neural level sets

---

### Meta-Review · Area_Chair_PgEq · 2026-01-04

**Summary:**

This paper presents a theoretical analysis of floating-point induced relative errors in cross-entropy (CE)-based gradient adversarial attacks, decomposing the problem into four attack regimes and deriving scenario-dependent optimal logit scaling factors. The work also proposes T-MIFPE, a theoretically motivated refinement of the existing MIFPE method, and provides experimental validation on standard robustness benchmarks.

Reviewers generally agreed that the paper is technically sound and theoretically careful, but raised consistent concerns regarding limited novelty beyond prior work, very small empirical impact, and clarity and accessibility of the presentation. While the rebuttal addressed several technical questions and strengthened the justification of assumptions, the core concerns about incremental contribution and practical significance remain.

**Reviewer Concerns:**

**Concerns addressed by the rebuttal:**

* **Clarification of theoretical assumptions**:
  The authors provided additional explanation and targeted experiments supporting the independence assumptions used in the analysis.
* **Experimental stability and rigor**:
  Additional multi-seed experiments, runtime overhead analysis, and integration with APGD-CE/AutoAttack improved the empirical completeness.
* **Presentation issues**:
  Some notation and template problems were corrected, and definitions were clarified in the revised version.

**Concerns that remain outstanding:**

* **Limited novelty relative to prior work (MIFPE)**:
  Despite the theoretical framing, multiple reviewers noted that the paper primarily formalizes and marginally refines an existing method whose core idea and empirical behavior were already established. The contribution is perceived as incremental rather than substantially new.
* **Very small empirical gains**:
  The observed improvements (often \~0.01-0.1%, occasionally up to \~0.3%) are consistently small. While theoretically explained, reviewers questioned whether this level of improvement justifies acceptance at ICLR.
* **Practical impact and scope**:
  The analysis is restricted to CE-based attacks, and the results do not significantly change the practical conclusions of robustness evaluations in most settings.
* **Overall readability and accessibility**:
  Even after revision, the paper remains mathematically dense and difficult to follow, limiting its accessibility to a broader ICLR audience.

**Reviewer Scores:**

* **Reviewer 1 (marginal reject)**:
  Likely unchanged; core concerns about novelty and impact remain.
* **Reviewer 2 (marginal reject, open to improve)**:
  May slightly improve, but still likely remains borderline.
* **Reviewer 3 (accept)**:
  Likely unchanged.
* **Reviewer 4 (marginal reject)**:
  Some concerns addressed, but remaining doubts about significance likely persist.

---

### Decision · Program_Chairs · 2026-01-26

Reject